# *HOP1* and *HAP2* are conserved components of the meiosis-related machinery required for successful mating in *Leishmania*

Carolina Moura Costa Catta-Preta[1], Tiago Rodrigues Ferreira [1],
Kashinath Ghosh[1], Andrea Paun[1] & David Sacks [1] ✉

Whole genome analysis of Leishmania hybrids generated experimentally in sand flies supports a meiotic mechanism of genetic exchange, with Mendelian segregation of the nuclear genome. Here, we perform functional analyses through the generation of double drug-resistant hybrids in vitro and in vivo (during sand fly infections) to assess the importance of conserved meiosis-related genes in recombination and plasmogamy. We report that HOP1 and a HAP2-paralog (HAP2-2) are essential components of the Leishmania meiosis machinery and cell-to-cell fusion mechanism, respectively, since deletion of either gene in one or both parents significantly reduces or completely abrogates mating competence. These findings significantly advance our understanding of sexual reproduction in Leishmania, with likely relevance to other trypanosomatids, by formally demonstrating the involvement of a meiotic protein homolog and a distinct fusogen that mediates non-canonical, bilateral fusion in the hybridizing cells.

For parasites of the *Leishmania* genus, genetic exchange is thought to contribute to the extraordinary diversity of species and strains which can produce a spectrum of diseases in their mammalian hosts, ranging from self-limiting cutaneous lesions to fatal forms of visceral disease. *Leishmania* parasites have a dimorphic life cycle, reproducing asexually as extracellular promastigotes within the alimentary tract of their sand fly vectors, and as intracellular amastigotes within the phagolysosomes of their mammalian host mononuclear phagocytes. Genetic exchange has been inferred from the analyses of genotypes observed in natural isolates, providing evidence for outcrossing and inbreeding[1–6]. In some instances the hybrid genomes have been associated with phenotype changes within their host environments, including increased transmission potential of infected sand flies and altered tissue tropism in infected humans[7,8].

Genetic exchange has been directly demonstrated by the generation of hybrids between different *Leishmania* strains and species, as well as between the same clone (self-hybridization), in the laboratory[9–14]. Experimentally, genetic exchange in *Leishmania* is non-obligatory, relatively rare, and thought to be naturally confined to life-cycle stages present in the sand fly midgut, although hybridization using promastigote and amastigote culture forms in vitro has recently been described[15–17]. Based on whole genome sequencing analyses, the allele inheritance patterns of experimental hybrids generated in sand flies provide strong evidence that the system of genetic exchange in *Leishmania* is Mendelian and involves meiosis-like sexual recombination[18]. This conclusion is reinforced by the identification of *Leishmania* homologs for meiotic genes[19], and their expression by promastigote stages recovered from sand flies[20], including the core meiotic genes *SPO11, HOP1* and *DMC1*, involved in creating DNA double-strand breaks, homologous chromosome alignment and recombination, respectively. Upregulated expression of the *Leishmania* homolog gene encoding the cell fusion protein HAP2/GCS1 (HAP-LESS 2/Generative Cell-Specific 1) has also been described for promastigotes able to hybridize in vitro[16]. Critically, haploid gametes have never been identified in sand flies or in vitro, and the presence and expression of these homologs does not necessarily mean that they function in meiosis, as expression of meiotic genes in some organisms can be detected in asexually growing cultures, e.g. *Acanthamoeba*[21].

---

[1]Laboratory of Parasitic Diseases, National Institute of Allergy and Infectious Diseases, National Institutes of Health, Bethesda, MD, USA.
✉e-mail: dsacks@niaid.nih.gov

The homologs may have also acquired non-meiotic functions, e.g., *SPO11* paralogs required for vegetative growth in *Arabidopsis*[22]. Furthermore, the fact that some sexually reproducing organisms, e.g. *Drosophila melanogaster*, lack many meiotic homologs, including *HOP1*, *HOP2*, *MND1*, *DMC1*, means that many core meiotic genes are not required for sex. Similarly, *HAP2*, which is an ancestral gamete fusogen required for sexual reproduction in major eukaryotic taxa, including protozoa, green algae, and flowering plants[23,24], is missing from other sexual taxa, including Fungi and most Metazoa[25]. Lastly, even if *Leishmania* express meiotic homologs, aneuploidy, which is a constitutive feature of *Leishmania* genomes, may disrupt normal meiotic processes, producing sterile gametes with unbalanced chromosome complements, as observed for the effects of polyploidization in some flowering plants[26].

The linkage of genes in the "meiosis toolkit"[27] to their functional roles in a meiotic process has not been formally validated for any parasites in the *Trypanosoma* and *Leishmania* genera. This includes the subspecies of *Trypanosoma brucei*, the agents of African trypanosomiasis, for which haploid gametes expressing meiotic gene homologs, including *MND1*, *DMC1*, *HOP1* and *HAP2*, have been observed in the salivary glands of the tsetse fly vector where mating is believed to take place[28–30]. In the current studies, we employed a functional genomics approach to identify genes required for *Leishmania* mating in vitro and in vivo. Using CRISPR/Cas9 editing tools to generate null mutants of core meiosis gene homologs (*HOP1*, *SPO11*, *DMC1*, *HOP2* and *MND1*), as well as two genes related to the HAP2 fusogen (here named *HAP2-1* and *HAP2-2*), we identified HOP1, a synaptonemal complex (SC) protein involved in chromosome pairing, and HAP2-2, as essential to *Leishmania* mating in vitro and in the sand fly. Complementation with the targeted genes could in each case partially rescue mating competence, and reporter constructs permitted detection of subpopulations of expressing cells and their localization to midgut microenvironments.

## Results

### *Leishmania* meiotic genes are not required for promastigote survival in vitro

We selected for our analysis seven genes previously identified as homologs of meiotic genes[16], including components of the synaptonemal complex (*HOP1*, *SPO11*, *DMC1*, *HOP2* and *MND1*), and two membrane proteins containing domains related to gamete fusion in several eukaryotes (here named HAP2-1 and HAP2-2). Using InterPro[31], we confirmed the main expected conserved domains for all genes of interest (Fig. 1A). HOP1 (LTRL590_360017200), while bearing the expected N-terminal HORMA (HOP1–REV7–MAD2) domain, does not possess a C-terminal Zn-finger motif required for DNA interaction in *Saccharomyces cerevisiae*[32], a characteristic shared with other organisms (Fig. 1B). HOP2 and MND1 harbor the expected DNA binding domains as WH-like (Winged Helix) and HTH (helix-turn-helix) motifs, respectively. Nevertheless, neither MND1 nor HOP2 feature the Leucine zipper domain (LZ3wCH), which is observed in *Chlamydomonas reinhardtii* for both MND1 and HOP2, and only in MND1 for *T. brucei* (Fig. 1B). SPO11 also contains a WH-like domain, as well as the

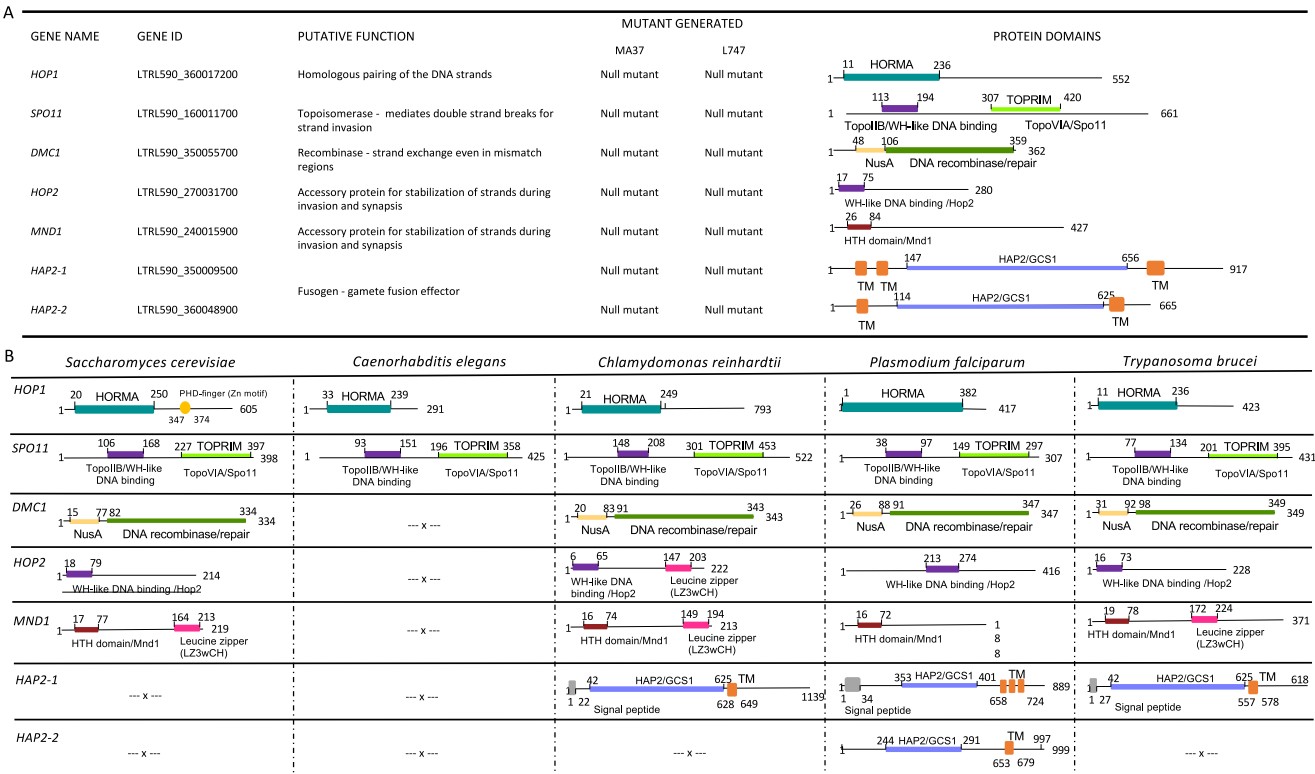

**Fig. 1 | The *L. tropica* genome encodes 5 synaptonemal complex related proteins and 2 plasma membrane fusogens. A** Gene names and gene IDs are presented for *L. tropica* reference genome L590 along with their putative functions based on model organisms for which meiotic proteins have been characterized. Null mutants were generated in MA37 and L747 *L. tropica* strains for all genes of interest. **B** Protein domains for meiosis-related genes in model organisms. Conserved protein domains of meiotic related proteins according to InterProScan {PMID: 24451626} are presented with the corresponding amino acid positions. HORMA domain, shared by a family of proteins (HOP1, REV7 and MAD2) that associate with axial elements; TopoIIB (topoisomerase IIB) domain, found in proteins which are able to induce double strand breaks in DNA; WH-like (Winged-Helix) domain with the 'helix' making sequence-specific DNA contacts with the major groove of the DNA and the 'wings' making contacts with the minor groove or backbone structure; TOPRIM (topoisomerase-primase) domain, which catalyzes the formation or cleavage of phosphodiester bonds; TopoVIA shares structural similarities with type IIB topoisomerase; NusA, transcription factor; HTH, Helix-Turn-Helix, structural DNA binding motif; Leucine zipper domain (LZ3wCH), provides the interface for heterodimer formation and interaction with DMC1; TM, transmembrane domain; HAP2/GCS2, fusogen domain present in transmembrane proteins found in eukaryotic male gametes or mating types.

TOPOPRIM/Topo6A domain common to type II topoisomerase VIA and SPO11. DMC1 is predicted to contain a N-terminal NusA domain, also found in other DNA repair proteins such as RAD51, as well as the conserved domain responsible for DNA recombinase and repair. Both proteins followed the well conserved structure and organization observed for other organisms (Fig. 1B). According to Tritrypdb searches[33], among members of the family, only organisms from *Leishmaniinae* and *Strigomonadinae* subfamilies encode two HAP2 proteins. In *L. tropica*, the two proteins share 26.36% of identity within the GCS1 domain. While HAP2-1 presents three predicted transmembrane domains (residues 50-73; 103-123; 663-686), HAP2-2 only has two (residues 50-73; 629-652). The signal peptide located at the N-terminus of HAP2 proteins was notably absent in both *Leishmania* HAP2-1 and HAP2-2. Additionally, the variability in the number of transmembrane (TM) domains is also observed in one of the HAP2 paralogues of *P. falciparum* (Fig. 1B). While *Leishmania* HAP2-1 is predicted to possess 2 TM domains at its N-terminus, HAP2-2 exhibits 1 TM domain, in addition to the one found at the C-terminus, a pattern shared with other organisms (Fig. 1A).

## Plasmogamy and synaptonemal complex related genes are involved in hybridization in vitro

To investigate the role of meiotic proteins in *Leishmania* genetic exchange, we targeted the 7 selected genes for deletion in *L. tropica* MA37 and L747, two strains that have previously been shown to yield a high frequency of hybrids recovery in co-infected sand flies[18] and in recent hybridization protocols in vitro[15,16]. For gene targeting, we used as background MA37 and L747 T7Cas9 cell lines[16], which stably express SpCas9 and T7RNA polymerase to drive single guide RNA (sgRNA) transcription[34].

For the design of sgRNA and recombination regions to generate the null mutants, we considered strain specific SNPs (Single Nucleotide Polymorphisms) to ensure correct targeting and gene deletion. We were able to obtain null mutants clones for each of the 7 genes on both strains (Supplementary Fig. 1A–O). For cell lines confirmation we performed two rounds of PCRs, the first to validate the removal the gene of interest (GOI), and the second to confirm the integration of the Puromycin-N-acetyltransferase gene (PAC) resistance gene within the targeted GOI locus. Details of the primers and plasmids for generation and confirmation of the mutant cell lines are provided in the Supplementary Data 4.

The null mutants were assessed for defects in promastigote proliferation in culture, and also tested for the effect of γ-irradiation, used to promote their in vitro hybridization, on promastigote survival. We did not observe any statistically significant changes in promastigote growth when comparing null mutants to their parental cell lines, nor in untreated and irradiated mutant lines (Supplementary Fig. 2).

To facilitate the hybrid screening, we generated fluorescent cell lines in each of the null mutants and T7Cas9 background controls. The resulting transfectants constitutively express either eGFP-Neomycin (Neo) resistance in MA37 strain or mCherry-Nourseothricin (Sat) resistance in L747 strain, integrated into the 18 S rDNA locus (Supplementary Fig. 1P–Q). To test which meiotic genes, if any, are required for genetic exchange in *Leishmania*, we performed a series of crosses in vitro to assess hybridization frequencies when only one or both parental lines were deleted of the gene of interest (details in Table 1 and Supplementary Data 1). The frequency of hybridization was compared to the rates obtained in MA37 eGFP-Neo and L747 mCherry-Sat control crosses. Since the parental lines from MA37 and L747 strains expressed eGFP and mCherry fluorescent proteins, respectively, all positive wells showing growth in the double drug selection medium were confirmed to be GFP⁺ mCherry⁺cells in our flow cytometry analysis (Fig. 2A). Four of the seven crosses performed when both parents were deleted of the gene of interest showed a significant decrease in

the percentage of positive wells (eGFP⁺ mCherry⁺), namely *HOP1* ($p < 0.0001$), *HOP2* ($p = 0.026$), *HAP2-1* ($p = 0.0003$) and *HAP2-2* ($p < 0.0001$), (Fig. 2B). Interestingly, absence of the fusogen-like protein HAP2-2 in either MA37 or L747 alone, was sufficient to reduce hybridization frequencies in vitro (Fig. 2B, C). For crosses involving the *HOP1* null mutants, deletion of the gene only in the L747 parent was sufficient to profoundly impair hybridization, while deletion in only the MA37 had a less pronounced but still significant inhibitory effect. Considering the initial number of cells in each co-culture to calculate the minimum frequency of mating competent cells, for *HAP2-1* we observed significant reduction only when the gene was lacking in both parental lines ($p = 0.005$), (Fig. 2C). For *HAP2-2*, by contrast, deletion either in MA37 or L747 alone was sufficient to reduce the frequency of mating competent cells (MA37 eGFP-Neo Δ*hap2-2*, $p = 0.002$; L747 mCherry-Sat Δ*hap2-2*, $p = 0.004$), while for *HOP1*, these frequencies were reduced when the gene was deleted in both parents ($p = 0.005$) or only L747 ($p = 0.02$).

To confirm that the hybridization defects were a direct result of deletion of the targeted genes, we used the original null mutant lines that did not express mCherry-Sat and re-expressed *HOP1* or *HAP2-2* by integrating the CDS plus the Sat resistance gene into the 18 S rDNA locus of each respective L747 null mutant, the strain showing the strongest hybridization defects (Fig. 2D). This allowed us to then compare the hybridization phenotypes of null mutants transfected with mCherry-Sat versus *HOP1*-Sat or *HAP2-2*-Sat. The non-fluorescent L747 null mutants were confirmed to have the same strong hybridization defect as the fluorescent null mutants when tested with their gene-sufficient MA37 non-fluorescent partner (Supplementary Fig. 3). Re-expression of either *HOP1* (L747 Δ*hop1*::*HOP1*) or *HAP2-2* (L747 Δ*hap2-2*::*HAP2-2*) conferred significant, though partial, reconstitution of L747 mating competence in vitro when paired with MA37 control (eGFP-Neo) or MA37 null mutants (Δ*hap2-2* eGFP-Neo or Δ*hop1* eGFP-Neo), (Fig. 2E, F). The presence of both parental drug resistance markers on clones generated from the double drug resistant lines was confirmed by PCR (Supplementary Figs. 4, 5). The partial recovery of hybridizing compatibility with the MA37 *HAP2-2* null mutant line was surprising since the crosses undertaken using the *HAP2-2* null mutant and the L747 control line (Fig. 2B, C) indicated that HAP2-2 expression in both parents was required for hybridization in vitro.

## Genetic exchange in the sand fly requires *HOP1* and *HAP2-2*

While our in vitro hybridization protocol facilitates screening of mutants' phenotypes, genetic exchange is described to naturally occur only in the sand fly vector[9]. Therefore, we tested if the hybridization deficits observed for the *HOP1* and *HAP2-2* null mutants would be reproduced during infections in *Lutzomyia longipalpis*, a permissive vector for experimental *L. tropica* infections and for hybrids generation in vivo[18]. Each parental line was tested alone for its ability to colonize the insect midgut. None of the mutant lines showed significant differences in parasite numbers after 8 days of infection (Fig. 3A, C). The pairwise combinations of controls and *HOP1* and *HAP2-2* mutants previously tested in vitro were used for sand fly co-infections. After 8 days of infection, homogenized midguts from individual flies were placed in double drug selection medium, and promastigotes from each double drug resistant line were tested by flow cytometry. The frequency of flies yielding a double drug resistant and dual fluorescent hybrid line was determined for each pairwise combination of parental lines tested. In the experiments involving the *HOP1* mutants, hybrids could be recovered from 57.5% of the flies co-infected with the MA37 and L747 parental controls (MA37 eGFP-Neo and L747 mCherry-Sat) (Fig. 3B). Consistent with the in vitro data, no hybrids could be recovered when *HOP1* was deleted only in L747 or in both parental lines ($p < 0.0001$). Deletion of *HOP1* in only the MA37 parent allowed hybrid recovery in 14% of flies ($p < 0.0001$). *HOP1*

**Table 1 | *L. tropica* control, null mutants, and re-expressors used to analyze the essentiality of meiotic-related genes for hybridization in vitro**

| Parental strains | | Frequency of hybrid recovery: #positive wells [a]/#total wells[b] (%) | *p*-value[c] |
|---|---|---|---|
| Crosses with fluorescent null mutants (Fig. 2B) | | | |
| *L. tropica* MA37 eGFP-Neo | *L. tropica* L747 mCherry-Sat | 40/72 (55.5%) | - |
| *L. tropica* MA37 eGFP-Neo | *L. tropica* L747 Δ*hop1* mCherry-Sat | 4/72 (5.6%) | 0.0583 |
| *L. tropica* MA37 Δ*hop1* eGFP-Neo | *L. tropica* L747 mCherry-Sat | 21/72 (29.2%) | <0.0001 |
| *L. tropica* MA37 Δ*hop1* eGFP-Neo | *L. tropica* L747 Δ*hop1* mCherry-Sat | 2/72 (2.8%) | <0.0001 |
| *L. tropica* MA37 eGFP-Neo | *L. tropica* L747 Δ*spo11* mCherry-Sat | 37/72 (51.4%) | 0.9996 |
| *L. tropica* MA37 Δ*spo11* eGFP-Neo | *L. tropica* L747 mCherry-Sat | 42/72 (58.3%) | 0.9994 |
| *L. tropica* MA37 Δ*spo11* eGFP-Neo | *L. tropica* L747 Δ*spo11* mCherry-Sat | 39/72 (54.1%) | 0.9998 |
| *L. tropica* MA37 eGFP-Neo | *L. tropica* L747 Δ*mnd1* mCherry-Sat | 43/72 (59.7%) | 0.7219 |
| *L. tropica* MA37 Δ*mnd1* eGFP-Neo | *L. tropica* L747 mCherry-Sat | 50/72 (69.4%) | 0.9993 |
| *L. tropica* MA37 Δ*mnd1* eGFP-Neo | *L. tropica* L747 Δ*mnd1* mCherry-Sat | 40/72 (55.6%) | >0.9999 |
| *L. tropica* MA37 eGFP-Neo | *L. tropica* L747 Δ*hop2* mCherry-Sat | 24/72 (33.3%) | 0.3938 |
| *L. tropica* MA37 Δ*hop2* eGFP-Neo | *L. tropica* L747 mCherry-Sat | 27/72 (37.5%) | 0.165 |
| *L. tropica* MA37 Δ*hop2* eGFP-Neo | *L. tropica* L747 Δ*hop2* mCherry-Sat | 19/72 (26.4%) | 0.0267 |
| *L. tropica* MA37 eGFP-Neo | *L. tropica* L747 Δ*dmc1* mCherry-Sat | 24/72 (33.3%) | 0.8405 |
| *L. tropica* MA37 Δ*dmc1* eGFP-Neo | *L. tropica* L747 mCherry-Sat | 31/72 (43.1%) | 0.1663 |
| *L. tropica* MA37 Δ*dmc1* eGFP-Neo | *L. tropica* L747 Δ*dmc1* mCherry-Sat | 25/72 (34.7%) | 0.2271 |
| *L. tropica* MA37 eGFP-Neo | *L. tropica* L747 Δ*hap2-1* mCherry-Sat | 14/72 (19.4%) | 0.0018 |
| *L. tropica* MA37 Δ*hap2-1* eGFP-Neo | *L. tropica* L747 mCherry-Sat | 13/72 (18.0%) | 0.0029 |
| *L. tropica* MA37 Δ*hap2-1* eGFP-Neo | *L. tropica* L747 Δ*hap2-1* mCherry-Sat | 9/72 (12.5%) | 0.0003 |
| *L. tropica* MA37 eGFP-Neo | *L. tropica* L747 Δ*hap2-2* mCherry-Sat | 1/72 (1.4%) | <0.0001 |
| *L. tropica* MA37 Δ*hap2-2* eGFP-Neo | *L. tropica* L747 mCherry-Sat | 0/72 (0%) | <0.0001 |
| *L. tropica* MA37 Δ*hap2-2* eGFP-Neo | *L. tropica* L747 Δ*hap2-2* mCherry-Sat | 0/72 (0%) | <0.0001 |
| Crosses with non-fluorescent null mutants (Fig. S3B) | | | |
| *L. tropica* MA37 SSU-Bsd | *L. tropica* L747 SSU-Pac | 26/72 (36.1%) | - |
| *L. tropica* MA37 SSU-Bsd | *L. tropica* L747 Δ*hop1*-Pac | 0/72 (0%) | <0.0001 |
| *L. tropica* MA37 SSU-Bsd | *L. tropica* L747 Δ*hap2-2*-Pac | 5/72 (6.9%) | 0.0004 |
| Crosses with re-expressors (Fig. 2E) | | | |
| *L. tropica* MA37 eGFP-Neo | *L. tropica* L747 Δ*hop1::HOP1*-Sat | 32/72 (44.4%) | <0.0001 |
| *L. tropica* MA37 Δ*hop1* eGFP-Neo | *L. tropica* L747 Δ*hop1::HOP1*-Sat | 28/72 (38.8%) | <0.0001 |
| *L. tropica* MA37 eGFP-Neo | *L. tropica* L747 Δ*hap2-2::HAP2-2*-Sat | 21/72 (29.1%) | 0.0002 |
| *L. tropica* MA37 Δ*hap2-2* eGFP-Neo | *L. tropica* L747 Δ*hap2-2::HAP2-2*-Sat | 26/72 (36.1%) | <0.0001 |

[a]Wells considered positive when parasites grew in double drugged media (G418 and Nourseothricin) and expressed both eGFP and mCherry.
[b]Total positive wells calculated based on 3 independent experiments.
[c]*p*-values < 0.05 are considered significant. For null mutant's crosses, *p*-values were calculated relative to control crosses, and re-expressors with the relevant gene-deficient crosses. *p*-values determined by one-way ANOVA for multiple comparison of preselected pairs and a two-step step-up method of Benjamini, Krieger and Yekutieli to correct false-discovery.

re-expression in the L747 null mutant significantly reconstituted hybrid recovery when paired with MA37 eGFP-Neo ($p = 0.0012$), albeit with lower efficiency compared to controls (MA37 eGFP-Neo and L747 mCherry-Sat). *HOP1* re-expression in the L747 null mutant line was sufficient to reconstitute mating compatibility even when paired with the MA37 *HOP1* null mutant ($p = 0.0295$), and at frequencies comparable to MA37 null mutant mating with the L747-Sat control parent (Fig. 3B).

Similarly, *HAP2-2* deletion in either one or both parental lines significantly impaired hybridization compared to controls (51.5% hybrid⁺ flies), (Fig. 3D). When sand flies were co-infected with MA37 eGFP-Neo and L747 Δ*hap2-2* mCherry-Sat, or MA37 Δ*hap2-2* eGFP-Neo and L747 Δ*hap2-2* mCherry-Sat, no hybrids were recovered ($p < 0.0001$), while deletion only in MA37 allowed for hybrid recovery in 5% of flies ($p = 0.0002$). Re-expression of *HAP2-2* in L747 partially rescued mating competence when crossed with MA37 control ($p = 0.0133$), but not with MA37 Δ*hap2-2* eGFP-Neo (Fig. 3D), supporting the requirement for HAP2-2 expression on both hybridizing partners in vivo.

## *HOP1* or *HAP2-2* deletion affect hybrid ploidy and parental chromosomal contribution

Although hybridization competence was substantially decreased in *L. tropica* MA37 deleted for either *HOP1* or *HAP2-2*, the few hybrids recovered from in vivo crosses with the gene sufficient L747 parent suggested that these genes were not strictly required in the MA37 strain for hybridization to occur. To determine the genome alterations of the hybrids resulting from the absence of *HOP1* or *HAP2-2* in the MA37 parent, we first performed propidium iodide (PI) staining to estimate ploidy based on the total DNA content of each hybrid, using as references the diploid parental lines, as well as diploid and tetraploid hybrids generated previously. From the sand fly experiments involving the *HOP1* mutants, we tested hybrids generated in two independent experiments (Supplementary Data 2) and found that MA37 Δ*hop1* eGFP-Neo and L747 mCherry-Sat crosses did not generate any diploid hybrids, a significant decrease when compared to control crosses (MA37 eGFP-Neo and L747 mCherry-Sat, $p = 0.041$; Fig. 4D). The control hybrids were mainly diploid ($n = 30$), although triploid ($n = 16$) and tetraploid ($n = 1$) hybrids were

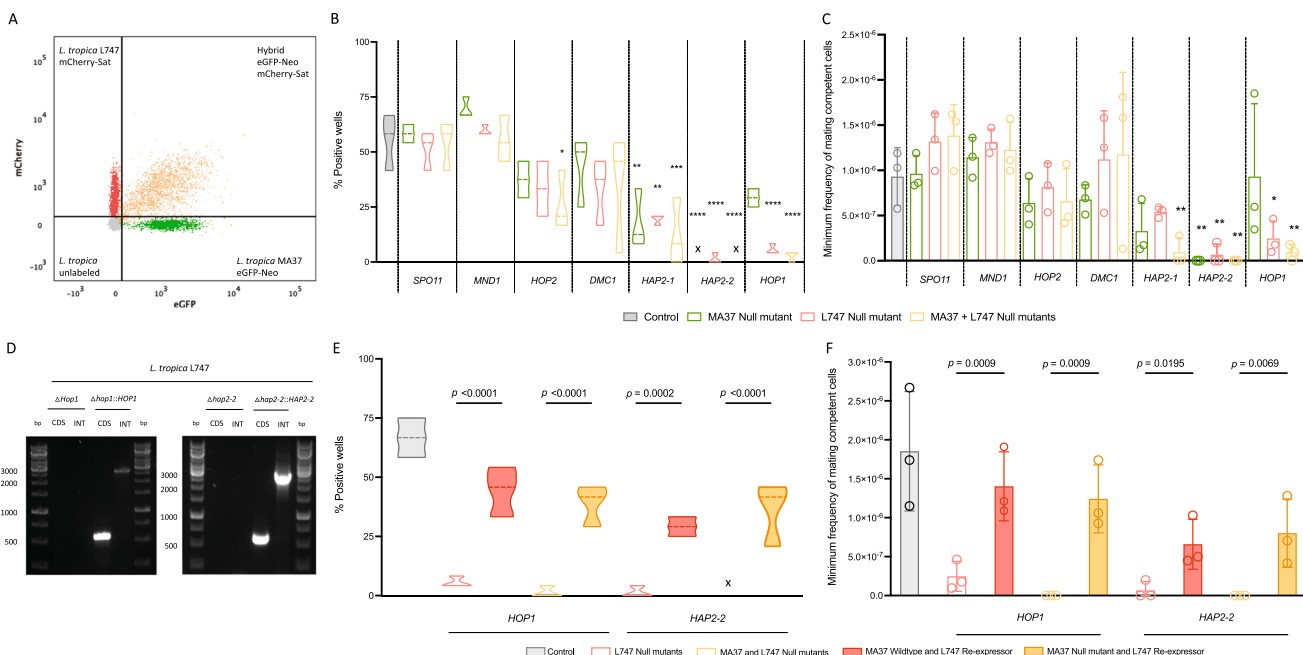

**Fig. 2 | Screening of parasite lines to identify meiosis-related genes required for in vitro hybridization in *L. tropica*. A** Representative flow cytometry analysis used for screening of double-fluorescent *L. tropica* hybrids. Non-fluorescent parasites (grey), parental cell line *L. tropica* MA37 eGFP-Neo (green), parental cell line *L. tropica* L747 mCherry-Sat (red) and hybrid expressing both GFP and mCherry (orange). **B** Violin plots for the percentage (%) of wells with cell growth in double-drug culture media (Neo and Sat). Control crosses (MA37 eGFP-Neo and L747 mCherry-Sat) are represented in grey, crosses with only one parent having the gene of interest deleted are in green (MA37) or red (L747), and crosses with both parental null mutant lines are in orange. All crosses were compared to the control crosses (grey): *$p = 0.0267$; **$p < 0.003$; ***$p \leq 0.0003$; ****$p < 0.0001$. See Table 1 for details of pairwise combinations of the parental lines used, percentages and total number of hybrids recovered, and precise *p*-values. **C** Bar graphs representing the minimum frequency of hybridization competent cells; colors of crossing combinations are the same as in (**B**). All crosses were compared to the control crosses (grey):

**$p \leq 0.0059$; *$p = 0.0225$. **D** Re-expressors were generated by integration of *HOP1* or *HAP2-2* genes at the small subunit RNA (SSU rRNA) locus. Cell lines were confirmed by PCR amplification of the CDS reintroduced in the null mutant cell line and the integration (INT) of the gene of interest and the drug resistance marker (Sat) in the SSU rRNA locus. bp, base pairs (**E**) Violin plots comparing the percentage of positive wells for hybrids from crosses performed with null mutants (*HOP1* or *HAP2-2*) in only L747 (red) or both parents (orange), and crosses performed with re-expressors in the L747 strain (filled violin plots). ***$p = 0.0002$; ****$p < 0.0001$. **F** Bar graphs representing the minimum frequency of mating competent cells for the crosses in E. ***$p \leq 0.0016$; *$p \leq 0.0195$. Results are represented as the mean of 3 independent experiments ±SD. *p*-values determined by one-way ANOVA for multiple comparison of preselected pairs and a two-step step-up method of Benjamini, Krieger and Yekutieli to correct false-discovery. Source data is provided in Supplementary Data 1.

also observed (Supplementary Fig. 6), similar to prior observations regarding the frequency of polyploid hybrids in vivo[9,11,18]. For the MA37 Δ*hop1* eGFP-Neo crosses, the hybrids were either triploids ($n = 2$) or tetraploids (Fig. 4A, D). Crosses between MA37 eGFP-Neo and L747 Δ*hop1::HOP1*, which confirmed rescue of the hybridization defect in the L747 Δ*hop1* parent, generated diploid ($n = 8$), triploid ($n = 2$) and tetraploid hybrids ($n = 7$), similar to control crosses (Fig. 4B, D). However, re-expression of *HOP1* in the L747 Δ*hop1* strain and crossing with MA37 Δ*hop1* eGFP-Neo parental line still generated only triploid ($n = 2$) and tetraploid hybrids ($n = 4$, $p = 0.027$; Fig. 4C), providing additional evidence that while the *HOP1* deleted line retains some ability to hybridize, it can only produce polyploid progeny. To test if the polyploid hybrids resulted from a possible defect in meiotic reduction in the deleted line, we used whole genome sequencing (WGS) analysis of the parents and individual triploid hybrids to estimate the genomic contribution from each parent. This analysis confirmed that the two triploid hybrids were full genome hybrids, with biallelic inheritance of all the SNPs that were homozygous and different between the parents (Fig. 4E). The circos plot of parental SNP contributions also shows that the triploid hybrids each inherited these alleles in a roughly 2:1 proportion (MA37:L747), confirming that the *HOP1* deficient parent was the source of the extra genome. For comparison, one diploid and one triploid hybrid from crosses involving the control parental lines are also shown, with the diploid hybrid showing balanced contributions from each parent and the triploid hybrid showing an extra genome complement from L747.

The parental SNP contributions averaged across individual chromosomes are shown in Supplementary Data 3.

Similar to the effects on polyploidy observed in the hybrids generated using the MA37 *HOP1* deficient strain, the MA37 Δ*hap2-2* eGFP-Neo strain also failed to generate diploid progeny ($n = 9$ hybrids), while 38 of the 49 hybrids generated between the parental controls were diploid ($p = 0.0035$; Fig. 5A, C). For the hybrids that were generated between L747 Δ*hap2-2::HAP2-2* and MA37 eGFP-Neo, mainly triploid ($n = 11$) and tetraploid hybrids ($n = 10$) were observed, with a significant decrease in diploids when compared to controls ($n = 2$; $p = 0.0082$; Fig. 5B, C). Thus, while *HAP2-2* re-expression in the deleted line rescued mating competence, it did not fully reconstitute either the frequency of mating or the ability to generate diploid offspring. Three of the triploid hybrids recovered from the crosses between MA37 Δ*hap2-2* and L747 mCherry-Sat were submitted for WGS, which revealed genome wide, biparental inheritance of the SNPs that were homozygous and different between the parents (Fig. 5D). The circos plot again reveals asymmetric parental contributions, although in this case the extra genome was contributed in all 3 hybrids by the *HAP2-2* sufficient parent and not the deleted cell line (Fig. 5D; Supplementary Data 3). The plot also reveals that a few chromosomes (e.g. LtHyb5, chr 4; LtHyb4, chrs 4 and 19) appeared to be inherited uniparentally, a phenomenon that we have observed previously with experimental hybrids and which we interpret as a loss of heterozygosity subsequent to the initial hybridizing event, associated with aneuploidy mechanisms that are known to operate during clonal growth in culture[35].

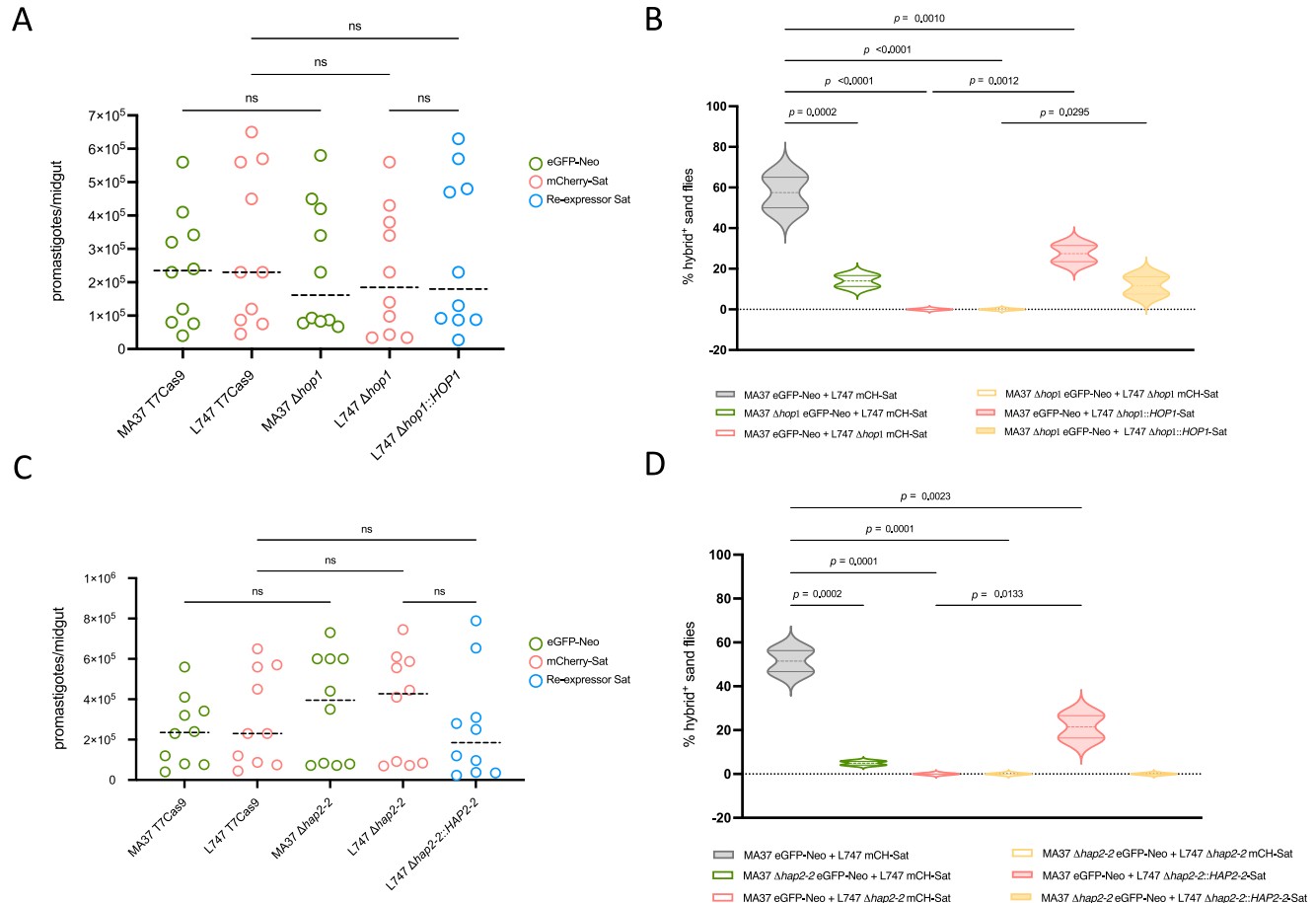

**Fig. 3 | Effects of *L. tropica HOP1* or *HAP2-2* deletion on *Lu. longipalpis* infection and parasite mating competence in vivo. A** Total number of promastigotes per midgut dissected from flies infected with controls, *HOP1* null mutants, or L747 *HOP1* re-expressor, 8 days post-infection, with each group consisting of 10 infected sand flies. Median is represented by the dashed line. **B** Percentage of sand fly midguts from which a *L. tropica* hybrid could be recovered (eGFP⁺ and mCherry⁺, Neo and Sat double resistant) in crosses using combinations of control cell lines (MA37 eGFP-Neo and L747 mCherry-Sat), *HOP1* null mutants and L747 *HOP1* re-expressor. **C** Total number of promastigotes per midgut dissected from flies infected with controls, *HAP2-2* null mutants, or L747 *HAP2-2* re-expressor, 8 days post-infection, with each group consisting of 10 infected sand flies. Median is represented by the dashed line. **D** Percentage of sand flies positive for a hybrid in crosses using combinations of control cell lines, *HAP2-2* null mutants and L747 *HAP2-2* re-expressor. Results are represented as the mean of 3 independent experiments ±SD. *p*-values determined by one-way ANOVA for multiple comparison of preselected pairs and a two-step step-up method of Benjamini, Krieger and Yekutieli to correct false-discovery. Source data is provided in Supplementary Data 2.

## HOP1 and HAP2-2 are expressed in promastigotes subpopulations during sand fly infections

Previous single-cell RNA-seq analysis indicated that HOP1 and HAP2-2 average expression is low and detectable in less than 10% of the population in culture either before or after γ-irradiation[16]. In order to visualize the promastigotes expressing these proteins during their development in the sand fly midgut, we generated reporter cell lines by inserting a fluorescent mNeonGreen (mNG) tag at the N-terminus of HOP1 and HAP2-2 in both MA37 and L747 non-fluorescent T7Cas9 strains. After confirming the correct integration in cultured cells (Supplementary Fig. 7), we infected *Lu. longipalpis* with the endogenously tagged mutants and dissected midguts at day 5 post infection. The protein localization in *L. tropica* was consistent for both MA37 and L747 strains, with HOP1 detected in a localized spot within the nucleus, while HAP2-2 was expressed throughout the cell body, including the flagellum (Figs. 6, 7). HAP2-2 could also be detected in an organelle localized either anterior or posterior to the nucleus, likely related to intracellular transport to the plasma membrane (Fig. 7A, C). Expression in each case was confined to a subpopulation of promastigotes, with HOP1 being expressed in much lower levels and in fewer cells than observed for HAP2-2. Cells expressing HOP1 could be found in the abdominal (Fig. 6A), but not in the thoracic midgut (Supplementary

Movie 1), while HAP2-2⁺ cells were also found in the thoracic midgut, imbedded within the promastigote secretory gel, a component of the plug (Fig. 7C). HOP1 was expressed in cells with 1 nucleus and 1 kinetoplast, and while the MA37 mNG::HOP1⁺ cells appeared larger than the HOP1⁻ cells, they did not otherwise show signs of cell division (Fig. 6A, arrow heads in DIC). HAP2-2⁺ cells were predominantly elongated or rounded cells with free elongated flagella (Supplementary Movie 2).

## Discussion

For several decades, genetic exchange in *Leishmania* has been implicated based on the description of natural hybrid genotypes[1,2,4–6,36], and later confirmed by the generation of hybrids in laboratory crosses in colonized sand flies[9]. Experimentally, genetic exchange in *Leishmania* remains a cryptic feature of its life cycle in the vector, and mating competent cells, and in particular, haploid gametic stages, have yet to be identified. Canonical meiosis in diploid organisms refers to the process in which a single round of chromosome duplication is followed by two rounds of reductional division to produce gametes which receive only a single copy of each homolog. The proper segregation of homologous chromosomes during prophase of meiosis I requires that the chromosomes undergo pairing, recombination, and synapsis[37]. The physical

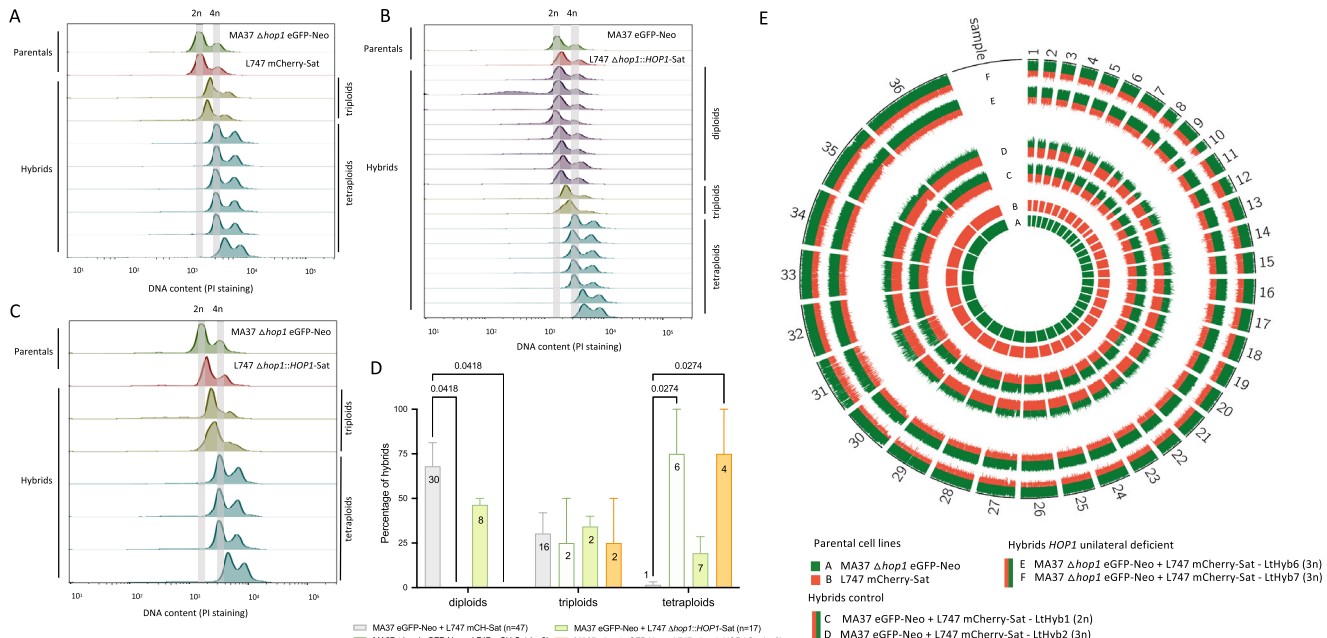

**Fig. 4 | Effect of *HOP1* deletion on ploidy and parental contributions of hybrids recovered from sand flies.** DNA content analysis by propidium iodide (PI) staining and flow cytometry of parents and hybrids from (**A**) MA37 Δ*hop1* eGFP-Neo and L747 mCherry-Sat; (**B**) MA37 eGFP-Neo and L747 Δ*hop1::HOP1*-Sat; and (**C**) MA37 Δ*hop1* eGFP-Neo and L747 Δ*hop1::HOP1*-Sat. The grey bands indicate the DNA content for 2n (G1/G0 peak) and 4n (G2/M peak) of the parents. **D** Quantification of the percentage of hybrids close to diploid (2n), triploid (3n) or tetraploid (4n) in control crosses (grey), MA37 Δ*hop1* eGFP-Neo and L747 mCherry-Sat (green line, empty bar), MA37 eGFP-Neo and L747 Δ*hop1::HOP1*-Sat (green filled bar), and MA37 Δ*hop1* eGFP-Neo and L747 Δ*hop1::HOP1*-Sat (orange filled bar). Numbers in the bars represent the total numbers of hybrids generated in the respective crosses from 2 independent experiments. In the image key, 'n' denotes the total number of hybrids for each crossing. Histograms of PI staining of control crosses can be found in Supplementary Fig. 6. *P*-values determined by one-way ANOVA for multiple comparison of preselected pairs and a two-step step-up method of Benjamini, Krieger and Yekutieli to correct false-discovery. **E** Circos plot representation of SNP frequencies in control and MA37 *HOP1* deficient crosses. Each block of the circle is labeled with its corresponding chromosome number. The colors represent the parental contribution of either MA37 (green; track A) or L747 (red; track B). SNP frequencies of a representative diploid (C) and triploid (D) hybrid from control crosses, and the two triploid hybrids recovered from MA37 *HOP1* deficient crosses (E, F) are shown. Allele frequencies are summarized in Supplementary Data 3. Analysis was performed using the software PAINT[56].

connections between the homologs occur in the context of a specific chromosomal structure, the synaptonemal complex (SC), and the main structural components of the SC, including cohesins and meiotic HORMA domain proteins, are evolutionarily conserved. The membrane proteins involved in the fusion of male and female gametes or mating types are also broadly conserved[38]. We show here evidence that the *Leishmania* homolog of HOP1, a meiotic core protein that functions in pairing of homologous chromosomes, and HAP2-2, a HAP2 family member involved in gamete fusion, are essential for genetic exchange in *Leishmania*.

HOP1 is a meiotic chromosome axis protein containing the HORMA domain, named after three proteins that harbor it (HOP1, REV7, and MAD2). HOP1 is known to perform meiosis specific functions in yeast (*Saccharomyces cerevisiae*), flowering plants (*Arabidopsis*), and worms (*Caenorhabditis elegans*)[39–41]. HORMA domain proteins are characterized in these and other organisms as essential for the recruitment of key components of the SC, which is a zipper-like multiprotein complex responsible for the synapsis, recombination, and segregation of homologous chromosomes at prophase of meiosis I[42]. The presence of HOP1 on the meiotic chromatin is described as a hotspot for the programmed DNA double strand break (DSB) machinery, and its displacement after SC establishment is associated with down-regulation of DSB activity[43]. Crossover formation during DSB repair, in addition to recombining genes along each chromosome, establishes the physical linkages required for accurate segregation of the homologs in meiosis I. In *Leishmania*, balanced segregation of the homologs in each parent is consistent with the whole genome sequencing analyses of large numbers of diploid hybrids recovered from sand flies, indicating that in each case the progeny have received

one full genome complement from each parent[18]. The severe mating defect displayed by the HOP1 null mutants in sand flies provides direct evidence that hybridization of wild type parasites is dependent on the SC meiotic machinery. While a few hybrids could be recovered when the MA37 *HOP1* null mutant was crossed with L747 wild type, the products were all polyploid, with the deleted parent shown to be the source of the extra genome in the triploid hybrids. Thus, HOP1 in MA37 is required for reductional division even if it is not absolutely essential for hybridization to occur. These results are similar to the outcome of the deletion of meiosis-specific genes in plants, for which defects in chromosome segregation produced hybrid progeny with aneuploid or polyploid genomes[44]. Still, since the hybridizing potential of the *HOP1* deleted lines was either strongly reduced or completely ablated, this would link a component of the meiotic machinery to the generation of fusogenic cells, and argue that the occasional polyploid hybrids generated in flies between wild type parents, observed in this and prior reports, are still products of fusion between cells that are committed to a meiotic program, even if they cannot complete meiotic division(s). This would explain the requirement for HOP1 in the generation of hybrids in vitro, which yields mainly tetraploid products, consistent with the fusion of diploid cells. The inability of the hybridizing cell to complete a meiotic program might be associated with the γ-irradiation treatment that was employed to facilitate hybrids generation in vitro, and that may produce DSBs that redirect DNA repair to a non-meiotic DNA damage response, including single-strand annealing and microhomology-mediated end joining[45], that does not involve crossovers. The fact that HOP1 is still required for hybrid generation suggests that the hybridizing cells are marked as gamete-like, even if they are unable to undergo reductional division(s). Unreduced gamete

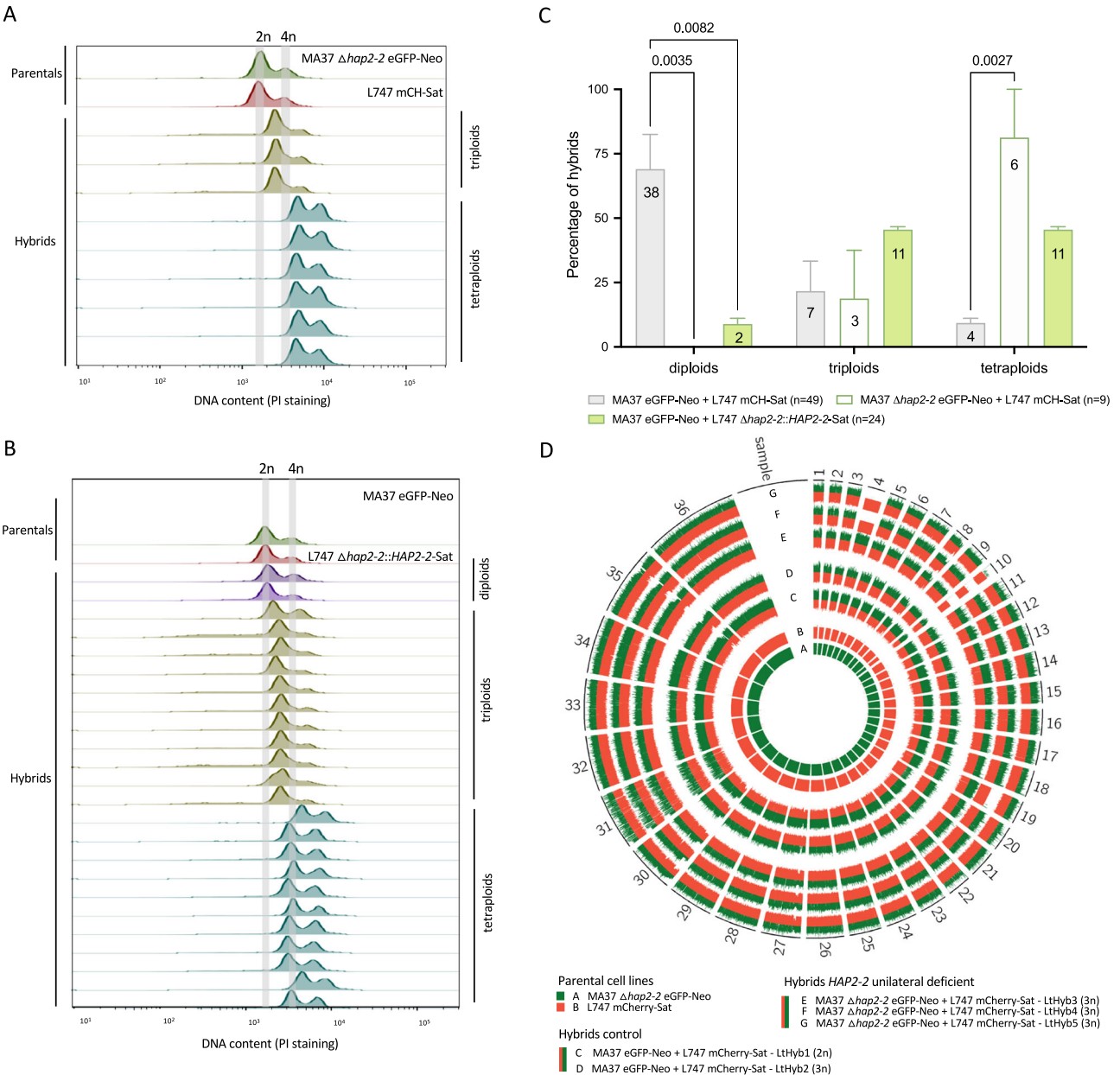

**Fig. 5 | Effect of *HAP2-2* deletion on ploidy and parental contribution of hybrids recovered from sand flies.** DNA content analysis by propidium iodide (PI) staining and flow cytometry of parents and hybrids from (**A**) MA37 Δ*hap2-2* eGFP-Neo and L747 mCherry-Sat; and (**B**) MA37 eGFP-Neo and L747 Δ*hap2-2::HAP2-2*-Sat. The grey bands indicate the DNA content for 2n (G1/G0 peak) and 4n (G2/M peak) of parents. **C** Quantification of the percentage of hybrids close to diploid (2n), triploid (3n) or tetraploid (4n) in control crosses (grey), MA37 Δ*hap2-2* eGFP-Neo and L747 mCherry-Sat (green line, empty bar), MA37 eGFP-Neo and L747 Δ*hap2-2::HAP2-2*-Sat (green filled bar). Numbers in the bars are the total numbers of hybrids generated in the respective crosses from 2 independent experiments. In the image key, 'n' denotes the total number of hybrids for each crossing. Histograms of PI staining

of control crosses can be found in Supplementary Fig. 6. *p*-values determined by one-way ANOVA for multiple comparison of preselected pairs and a two-step step-up method of Benjamini, Krieger and Yekutieli to correct false-discovery. **D** Circos plot representation of SNP frequencies in control and MA37 *HAP2-2* deficient crosses. Each block of the circle is labeled with its corresponding chromosome number. The colors represent the parental contribution of one parental line, either MA37 (green; track A) or L747 (red; track B). SNP frequencies of a representative diploid (C) and triploid (D) hybrid from control crosses, and the three triploid hybrids (E, F, & G) recovered from MA37 *HAP2-2* deficient crosses are shown. Allele frequencies are summarized in Supplementary Data 3. Analysis was performed using the software PAINT[56]. Source data is provided in Supplementary Data 3.

formation can occur naturally and is a common mechanism of polyploidization in plants[46].

We also investigated the importance of other meiotic gene homologs involved in early events of meiosis I, namely SPO11, involved in DSB formation, and DMC1 and its accessory proteins MND1 and HOP2, involved in DNA strand invasion and recombination. We could not associate any of these genes to hybridization in vitro. In contrast to the effects of HOP1 deletion, the early stages of gametogenesis may be able to proceed in the absence of these axial-elements, or else there are

proteins/pathways that provide redundant functions. In *C. elegans*, for example, radiation induced DSBs were shown to alleviate the requirement for SPO11[47]. The RecA homologs DMC1 and RAD51 serve complementary functions for strand exchange between homologous chromosomes in *S. cerevisiae*, although homology search and strand invasion can still occur with low efficiency in the absence of DMC1[48]. It will be important to extend the behavior of each of these null mutants to their mating competence in sand flies, in the context of what appears to be a complete meiotic cycle.

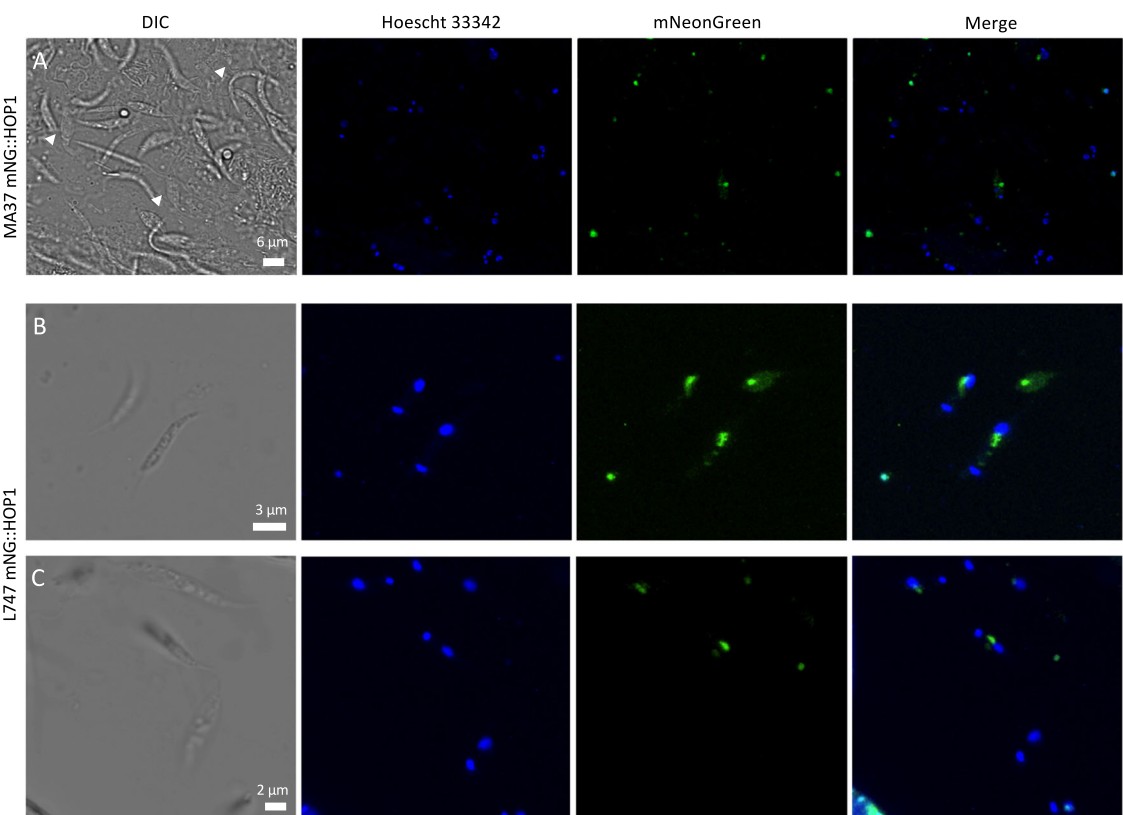

**Fig. 6 | Expression of HOP1 during *Lu. longipalpis* infection. A** Widefield differential interference contrast (DIC) image of *L. tropica* MA37 mNG::HOP1 promastigotes, released from a day 5-infected midgut. White arrowheads indicate *HOP1* expression in the nuclear region of larger cells containing 1 nucleus and 1 kinetoplast. **B, C** Widefield image of *L. tropica* L747 mNG::HOP1, showing expression of HOP1 in the nucleus of non-dividing cells. Parasites were released from the tissue for better detection of the punctate signal. Hoechst staining in blue, mNeonGreen in green. Scale bars are indicated. Infections and imaging were independently repeated at least twice, with the analysis of 3–5 midguts each time.

The cell fusion of male and female gametes culminates the classical meiotic program. The gene encoding HAP2 *(from HAPLESS2)*, synonymous with *GCS1 (Generative cell specific 1)*, was first described in *Arabidopsis* and is now known to catalyze zygote formation across major eukaryotic taxa, from protists to insects[49]. The slime mold *Dictyostelium discoideum*[50] and *Plasmodium spp.*[51] express two *HAP2*-related genes, both of which are critical for mating. We previously reported that *Leishmania* also possesses two *HAP2-like* encoding genes[16], designated here as *HAP2-1* and *HAP2-2*. In our prior study, positive selection of HAP2-1 expressing cells in one or both parents was associated with hybridization competence in vitro; association with HAP2-2 expression was not addressed[52]. The current studies directly address and clarify the roles of these fusogens in *Leishmania* hybridization in vitro. Deletion of *HAP2-1* in either or both parents produced a significant, though modest reduction in the number of in vitro hybrids recovered. When calculated as the minimum frequency of hybridizing cells in the co-cultures, a significant reduction was observed only when the gene was deleted in both parental lines. By contrast, deletion of *HAP2-2* in either or both parents produced a near complete hybridization defect in culture, a phenotype that was extended to sand flies. The data suggest that HAP2-1 expression is required on only one of the fusing cells, while HAP2-2 expression is required on both. It is possible that HAP2-2 function may have mitigated the effects of the HAP2-1 deletion, although the reverse does not seem to be the case, with the possible exception of the few hybrids that were generated between MA37 *HAP2-2* null mutant and L747 wildtype. The requirement for expression of HAP2-2 on each hybridizing partner is unusual, as other organisms require HAP2 only on one of the two gametes, such as the mating type minus gamete in *Chlamydomonas* algae, or the male

gametes in *Plasmodium*, *Tetrahymena*, or *Arabidopsis*, consistent with its functional similarity to fusion proteins of enveloped viruses[38,53–55]. By contrast, and similar to our observations regarding HAP2-2, fusion in *Dictyostelium* was completely blocked by single disruption of the *HAP2* locus, suggesting that bilateral interactions were needed to mediate membrane fusion[50]. It is interesting that the few hybrids that could be generated in flies between MA37 *HAP2-2* null mutant and L747 wildtype, were all either triploid or tetraploid, while the crosses between the wildtype lines were mainly diploid. HAP2-2 deletion did not appear to be directly linked to defective reductional division, as our WGS analysis of the 3 triploid hybrids determined that the extra genome was contributed in each case by the wild type parent, not the deleted line. It is possible that in contrast to HAP2-2 mediated fusion, the HAP2-1 dependent fusion that may operate following disruption of HAP2-2 is not confined to cells that are committed to a meiotic program. HAP2-1 was in fact shown to be constitutively expressed on a high frequency of cultured cells[52]. Future studies will explore the cooperative roles of the two fusogens during sexual mating in *Leishmania*.

HAP2 in *Tetrahymena* and *Chlamydomonas* is expressed on apically localized regions of the plasma membrane where the cells fuse[38,54]. For the *L. tropica* HAP2-2 reporter lines recovered from infected sand flies, we observed uniform expression throughout the cell body, including the flagellum, which might reflect an indiscriminate gamete attachment site, similar to gamete fusion in *Plasmodium*. In *P. falciparum*, HAP2 is expressed throughout the plasma membrane of male gametocytes beginning at stage II of development, and in fully differentiated microgametes. Its paralog, HAP2p, shows a patchier membrane expression pattern, and begins to be

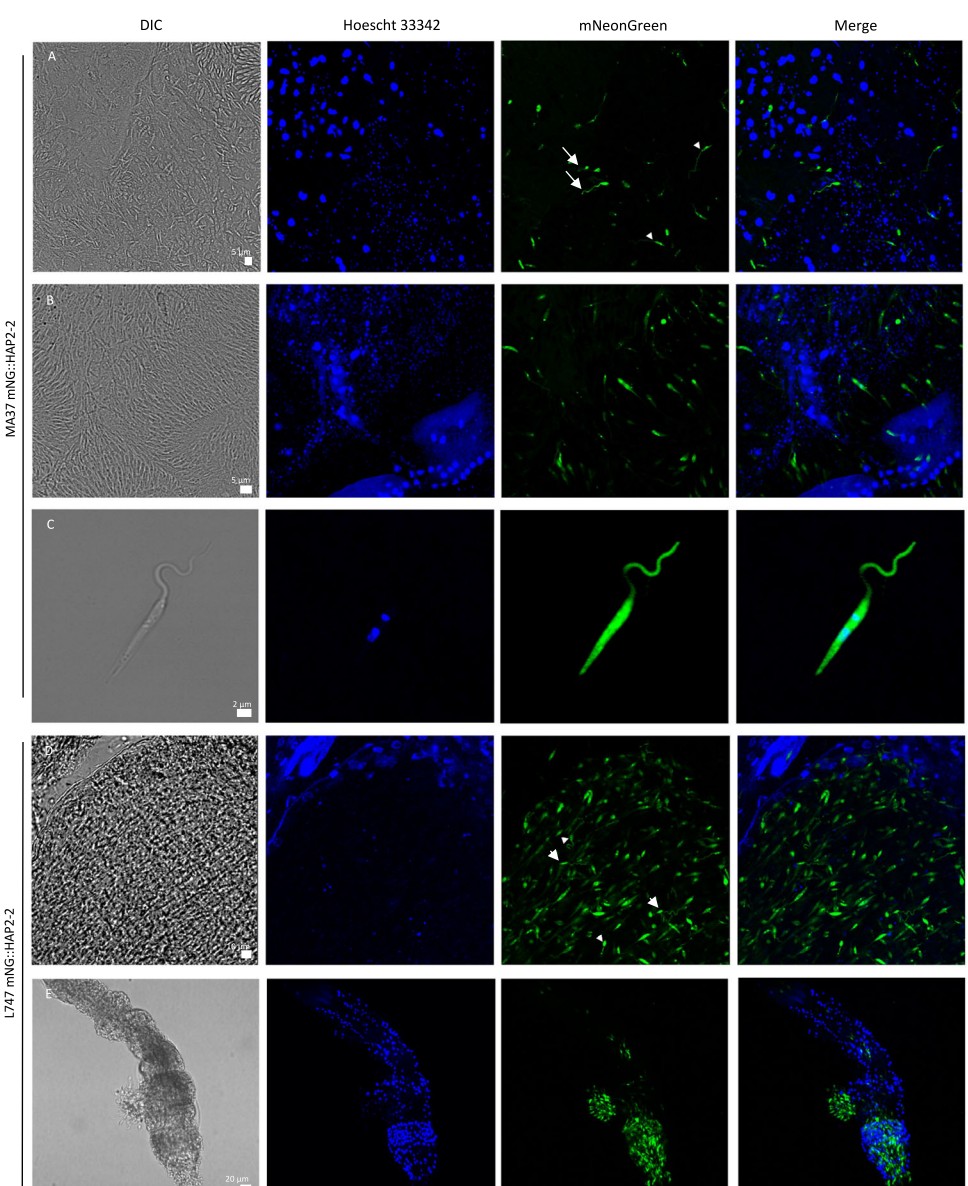

**Fig. 7 | Expression of HAP2-2 during *Lu. longipalpis* infection. A** Abdominal midgut tissue after 5 days of infection showing expression of HAP2-2 in a minority of MA37 mNG::HAP2-2 promastigotes. DNA staining (Hoescht 33342, blue) differentiates the larger nucleus of insect cells and the smaller nucleus of promastigotes. Arrowheads point to cells expressing HAP2-2 in the cytoplasm, and white arrows indicate smaller, rounded cells with free flagella expressing HAP2-2 on the surface. **B** Abdominal midgut showing a sub-population of promastigotes attached to the midgut wall expressing HAP2-2, with diffuse and/or highly localized staining in the cytoplasm. **C** Higher magnification of a promastigote released from the midgut showing diffuse expression of HAP2-2 throughout the cell body, including the flagellum. **D** Abdominal midgut tissue with a high proportion of promastigotes expressing L747 mNG::HAP2-2 in the cytoplasm. Arrowheads indicate rounded cells expressing HAP2-2 either in the cytoplasm or on the surface, with elongated forms expressing it in the cytoplasm (white arrows). **E** Thoracic midgut after 5 days of infection showing expression of HAP2-2 in a majority of MA37 mNG::HAP2-2 promastigotes. Hoechst staining in blue, mNeonGreen in green. Scale bars are indicated in the DIC images. Infections and imaging were independently repeated at least twice, with the analysis of 3–5 midguts each time.

expressed later during gametocyte development, and interestingly on stage V female gametocytes[51]. Whether HAP2p functions bilaterally to promote gamete fusion in *P. falciparum*, as described here for HAP2-2 in *L. tropica*, has not been addressed. In *T. brucei*, HAP2 expression is concentrated in some gametes at the cell posterior, while in other gametes and meiotic intermediates, it is expressed throughout the cytoplasm[30]. Whether it needs to be expressed on both gametes is not known. HAP2 expression in *T. brucei* is confined to life cycle stages in the tsetse salivary glands where gametes and meiotic intermediates are found. Similarly, HOP1 expression in *T. brucei* is restricted to cells in the salivary glands undergoing meiotic division, marked by the presence of two kinetoplasts[28]. By comparison, the HAP2-2 expressing cells in *L. tropica* were distributed throughout the posterior and anterior midgut, including the plug, while the HOP1 expressing cells were mainly in the posterior midgut and did not show signs of cell division. We so far lack evidence that HAP2-2 and/or HOP1 expression is restricted to gametes or cells committed to a meiotic program.

In summary, using reverse genetics and in vitro and in vivo protocols, we provide evidence that HOP1, a meiosis machinery protein, as well as a HAP2 fusogen family member, are essential for mating in *Leishmania*. As these are also the first functional studies of meiotic stage-related proteins in trypanosomatids, they are relevant to the mechanisms of meiotic recombination and plasmogamy in

other pathogenic species, including *Trypanosoma cruzi* and African trypanosomes.

## Methods

### Parasites

For this study we used two background cell lines previously reported: *L. tropica* MA37 T7Cas9-Hyg and *L. tropica* L747 T7Cas9-Hyg, generated from strains (MHOM/IL/02/LRC-747 and MHOM/JO/94/MA37) originating from Jordan and Israel, respectively[16]. Axenic cultures were maintained at 26 °C in cM199 (complete M199: 20% heat-inactivated Fetal Bovine Serum, 1% 10 mM Adenine, 0.2% 7 mM Hemin, 1% 100x Penicillin/Streptomycin, 1% 100x Glutamax, 0.4% 2 mM Biopterin, 0.1% 4 mM Biotin, 2.6% 10x M199 and 1x M199 q.s.p. 1 L). Cell lines were not kept in culture for more than 3 passages. For all experiments frozen stocks of cells were thawed and after 48-60 h recovery, cultures were diluted (1:100) in the presence of selective antibiotics.

For generation of in vitro hybrids, 5 mL of each of the parental cell lines ($5.0 \times 10^6$ mL$^{-1}$) were cultivated without selection pressure for 24 h before exposure to 6.5 Gy of γ-irradiation. The cell concentration was estimated before mixing together equal volumes of each parental line and distributing it in 96-well microplates (100 μL/well). Co-cultures were diluted (1:10) in double-drugged cM199 (Neomycin, Thermo Fisher - Neo 50 μg/mL and Nourseothricin, Jenna-Bioscience - Sat 300 μg/mL) after 24 h. For control crosses with non-fluorescent MA37 SSU-Bsd control parental line and L747 SSU-Pac, L747 Δ*hop1*-Pac or L747 Δ*hap2-2*-Pac, co-cultures were selected and maintained in cM199 with Blasticidin, Fisher Scientific (Bsd, 20 μg/mL) and Puromycin, Sigma-Aldrich (Pac, 40 μg/mL). Drug selection controls were performed in parallel with γ-irradiated parental lines submitted to pressure of selective antibiotics separately or combined. Selection plates were diluted weekly in fresh, double-drugged media and hybrids confirmed by the co-expression of fluorescent markers using flow cytometry (FACSCanto II and FACSDiva software {SCR_001456}). For crosses using non-fluorescent parental cell lines, hybrids were cloned and one clone from each well was confirmed by PCR for correct integration of the resistance markers, as described below for the confirmation of the parental lines (oligonucleotides details in the Supplementary Data 4). Independent experiments were performed in triplicate and the percentage of positive wells calculated. The minimum frequency of hybridization/mating competence was calculated assuming 1 hybridization event per well and using the number of input cells in L747 parental line and the number of positive wells in each cross $\left( \frac{positive\ wells}{(total\ number\ of\ wells \times initial\ cell\ concentration\ parental\ 1)} \right)$.

### Infection of sand flies

Infection of *Lu. longipalpis* sand flies with *L. tropica* MA37 and L747 controls, null mutants and re-expressors were performed as described by Inbar et al.[18]. Briefly, log-phase promastigotes ($5.0 \times 10^6$ mL$^{-1}$ total) were resuspended in heparinized mouse blood reconstituted with heat-inactivated serum and used to fed 2-4 days old female sand flies through a chick-skin membrane. Unfed sand flies were discarded after 24 h, and engorged females kept at 26 °C in a wet-chamber and provided 30% sucrose ad libitum. Sand fly midguts were dissected at day 5 for imaging, and at day 8 post-infection for counting and selection of hybrids.

The mice used in this study were used under a study protocol approved by the NIAID Animal Care and Use Committee (protocol number LPD 68E). All aspects of the use of mice in this research were monitored for compliance with The Animal Welfare Act, the PHS Policy, the U.S. Government Principles for the Utilization and Care of Vertebrate Animals Used in Testing, Research, and Training, and the National Institutes of Health (NIH) Guide for the Care and Use of Laboratory Animals. Housing conditions included a standard 12-hour dark/light cycle, with the ambient temperature maintained between 22–24 °C, and the humidity level set at 60%.

For single cell line infections, the total number of promastigotes was estimated on a hemocytometer. For recovery of hybrids, sand flies were infected with equal concentration of each of the parental cell lines, and after 8 days each midgut was homogenized and added to double drugged cM199 (Neomycin - Neo 50 μg/mL and Nourseothricin - Sat 300 μg/mL) containing 0.2% 50 mg/mL Gentamicin before plating into a single well of a 96-well microplate. Wells positive for promastigote growth after 7–10 days of culture were checked by flow cytometry (FACSCanto II and FACSDiva software) for co-expression of mCherry and GFP. Hybrids generated using the re-expressors were confirmed by PCR. Results were plotted as the percentage of sand flies yielding a hybrid.

### Bioinformatic identification of meiosis-related genes

Genomic and protein sequences were retrieved from the *L. tropica* L590 reference genome in TriTrypDB (https://tritrypdb.org/tritrypdb/) and the conserved domains identified using InterproScan searches. The percentage identity of HAP2-1 and HAP2-2 GCS domains were determined by alignment using MUSCLE (multiple sequence alignment with high accuracy and high throughput) {SCR_011812}.

### Generation of cell lines

For CRISPR-Cas9 mediated genomic modifications, MA37 T7Cas9 and L747 T7Cas9 background cell lines were transfected with repair cassettes and sgRNA linear DNA produced using protocols described in Beneke et al.[34]. Primers were designed using LeishGEdit (http://www.leishgedit.net/) based on the *L. tropica* L590 reference strain and manually corrected for SNPs of L747 and MA37 strains. A limit of 3 SNPs per sgRNA plus homologous recombination region was tolerated between the strains as long as they were not mapped to the PAM region. Repair cassettes produced to generate null and endogenously tagged mutants were generated using pTPuro_v1 and pPLOTv1 blast-mNeonGreen-blast as template, respectively.

To generate fluorescent and re-expressors cell lines in wildtype (T7Cas9) and null mutants cell lines, pLEXSY-cherry-Sat2, pLEXSY-LtroL747_HOP1-Sat2, pLEXSY-LtroL747_HAP2-2-Sat2, A2-GFP-Neo plasmids were SwaI-digested (Thermo Fisher) for linearization and purified from gel with QIAquick Gel Extraction Kit (QIAGEN) before transfection for integration in the small subunit rRNA (SSU).

Log-phase promastigotes ($1.0 \times 10^7$ mL$^{-1}$) were resuspended in P3 primary cell 4D buffer (Lonza), mixed with DNA fragments (total 1–2 μg) immediately before transferring to a Nucleofector strip and transfected using program FI-115 (Amaxa 4D-Nucleofector, Lonza). Cells were recovered in 5 mL of cM199 for 18 h before selection with antibiotics: Puromycin (Pac) - 40 μg/mL; Blasticidin (Bsd) - 20 μg/mL; Nourseothricin (Sat) - 300 μg/mL; Neomycin (Neo) - 50 μg/mL. Transfectants were immediately cloned by dilution (1:10 and 1:100) in cM199 with appropriate antibiotics before distributing 100 μL per well in 96-well microplates. Population and control selections were performed in 25 cm³ flasks.

Clones were recovered after 10–14 days for null mutants and 7–10 days for reporters, fluorescent and re-expressor cell lines. Outgrowths were split at least once in 10 mL of cM199 and suitable antibiotics. Genomic DNA was harvested from cultures using DNeasy Blood and Tissue kit (QIAGEN) and mutants were confirmed by PCR. For null mutants the presence/absence of the target gene was confirmed with primers that anneal inside the CDS, and the integration of the repair cassette in the gene locus confirmed by amplification from the 5′-UTR of the gene deleted until the 5′ end of the Pac resistance gene (CP-OL091). Integration of mNeonGreen-Bsd repair cassette was confirmed using a similar strategy but using a Bsd specific reverse primer (CP-OL062). To confirm fluorescent cell lines and re-expressors, integration in the SSU was assessed by combining a forward primer annealing to the resistance marker (Neo: CP-OL244; Sat:

CP-OL245) with a reverse primer annealing downstream of the recombination site in the SSU (CP-OL243). For re-expressors, PCR reactions using CDS specific primers were performed in parallel. Primer details are described in the Supplementary Data 4.

### DNA cloning
To generate re-expressor constructs, the backbone vector was prepared from pLEXSY-mCherry-Sat2 by double digestion with BglII and NotI (Thermo Fisher) and gel purification with QIAquick Gel Extraction Kit (QIAGEN). The CDS of L747 *HOP1* and L747 *HAP2-2* were amplified from *L. tropica* L747 T7Cas9 genomic DNA using primers designed with the NEBuilder assembly tool (details on Supplementary Data 4) that contains a 20 nt overlap sequence to the backbone of the vector. Construct assemblies were performed by incubating equimolar DNA concentrations of the insert and the digested backbone vector with NEBuilder HiFi DNA Assembly Master Mix (New England BioLabs) for 1 h at 50 °C. After bacterial transformation, colony PCR (Bioline) was performed using a forward primer binding at the backbone vector (CP-OL102) and a reverse primer annealing inside the insert (*HOP1*: CP-OL079; *HAP2-2*: CP-OL082). Constructs were confirmed by Sanger sequencing before preparing DNA for transfection.

### Flow cytometry and DNA content
Ploidy of parental lines and hybrids was estimated from propidium iodide (PI) stained mid-log phase promastigotes. Cells were incubated in 0.4% paraformaldehyde for 1 min at room temperature. After centrifugation, cells were resuspended in 100 μL of PBS, topped with 1 mL of cold 100% methanol and kept on ice for 15 min. Next, cells were sedimented and stained with PI and treated with RNase (13 mg/mL each, both from Sigma-Aldrich) at room temperature for 15 min. Cells were washed and resuspended in cold PBS and data collected by flow cytometry (FACSCanto II and FACSDiva software). DNA content analysis was performed using FlowJo software v.10.8 {SCR_008520} (Becton, Dickinson and Company) comparing parental lines and hybrids.

### Confocal microscopy
Midguts were dissected and transferred to the VectaVIEW (Vector) working solution (10 μL Reagent A: 10 μL Reagent B: 10 μL Reagent C) and incubated at room temperature for 10 min. Midguts were quickly transferred to PBS before incubation in 10 μg/mL of Hoechst 33342 (Thermo Fisher) for 20 min. After rinsing in PBS, the midguts were mounted on ice-chilled glass slides containing 10 μL of CyGEL Sustain (Abcam), covered with coverslips, and sealed to keep parasites immobile during imaging. Images were obtained using a 20 x /1.3 oil objective with the following acquisition mode: bidirectional scan, scan speed 600 Hz, line average 3 and sequential scan between stacks. We used the diode (405 nm) and the WLL (70%) lasers, with laser line set for Hoechst 33342 1% (410–450 nm) and mNeonGreen 5% (480–510 nm) on a Leica SP8 WLL FLIM confocal microscope in LAX software {SCR_013673} (Leica Microsystems). Projections and videos of the Z-axis were processed using Imaris 9.8.2 {SCR_007377} (Oxford Instruments).

### Whole genome DNA sequencing and parental SNP contribution analyses
Parental cell lines and sand fly derived hybrids were cloned by limiting dilution and their genomic DNA was extracted using DNeasy Blood and Tissue kit (QIAGEN) and submitted to Psomagen for next-generation sequencing (Rockville, MD). DNA libraries were generated using Tru-Seq Nano DNA Library Prep kit (Illumina) and the 150 bp-paired-end reads were sequenced on a NovaSeq6000 (Illumina).

Paired-end reads were aligned to the *L. tropica* Ltr590 v.57 reference genome available on TritrypDB (tritrypdb.org), using the BWA-MEM aligner v.0.7.17 with default parameters. Mean sequencing coverage of mapped reads was 30.10 (SD = 17.03), according to

Qualimap v.2.2.1. Single nucleotide polymorphisms (SNPs) were determined using the PAINT software suite designed for studying inheritance patterns in aneuploid genomes {PMID: 33530584}. PAINT was also used to find and extract the homozygous SNP marker differences between the parental strains and to estimate the chromosome copy numbers (somy). Chromosome somies were determined by calculating the normalized median read depth multiplied by 2 (for a diploid genome) using the *ConcatenatedPloidyMatrix* utility with a 5-kb window size. Genomic regions of multiple sequence repeats and high copy number variation (CNV) were filtered out from the analysis by eliminating positions with coverage levels ≥2-fold and ≤0.5-fold the average chromosome coverage. In the case of the polyploid hybrids (≥3n), the somy values were divided by 2 and multiplied by the ploidy estimated from the DNA content analysis (PI staining) and the parental contribution profile (e.g., 1:1 or 2:1 parental contribution ratio).

SNPs were considered heterozygous if the allele frequencies were between 0.15–0.85 and homozygous if >0.85 for each genomic position. Allele frequencies of <0.15, read depth lower than 10 or represented by <25% of reads in either forward or reverse direction were filtered out of the analysis. The allelic inheritance of each homozygous parental SNP in the hybrid progenies was determined using the *getParentAllelFrequencies* utility in PAINT. The parental allele frequencies were formatted to be compatible with Circos software v.0.69 {PMID: 19541911} and inheritance circos plots were generated with 372,866 homozygous marker differences between *L. tropica* MA37-GFP and L747-mCherry cell lines labeled in green and red, respectively.

### Statistical analyses
Hybridization experiments in vitro were independently repeated three times, while in vivo mating experiments were repeated twice. No randomization or blinding procedures were used in this study. For statistical analysis, data was plotted and analyzed on GraphPad Prism 9.3. We used Ordinary one-way ANOVA for multiple comparison of pre-selected pairs and a two-step step-up method of Benjamini, Krieger and Yekutieli to correct false discovery. *p*-values < 0.05 were considered significant.

### Reporting summary
Further information on research design is available in the Nature Portfolio Reporting Summary linked to this article.

## Data availability
The WGS data generated in this study have been deposited in the NCBI BioProject database under accession code PRJNA972461. The hybridization source data and WGS detailed analysis generated in this study are provided in Supplementary Data 1, 2 and 3.

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

## Acknowledgements

We thank Steve Beverley for sharing his insights regarding the presence of a HAP2 paralog in *Leishmania*. This work was supported in part by the Intramural Research Program of the NIAID, NIH.

## Author contributions

C.M.C.C.-P. designed, performed, and analyzed the experiments, and wrote the manuscript. T.R.F. designed in vitro hybridization experiments and performed bioinformatic analysis; K.G. was responsible for the maintenance of the insects; A.P. helped with parasite cultures and insect maintenance; D.S. designed the project and wrote the manuscript.

## Funding

## Competing interests

The authors declare no competing interests.
