## [Peer Review File · Nature Communications]

HOP1 and HAP2 are conserved components of the meiosis-related machinery required for successful mating in *Leishmania*Reviewers' Comments:

Reviewer #1:

Remarks to the Author:

The topic of sexual reproduction in *Leishmania* is an important one because this genus includes several human pathogens with devastating global impacts. Genetic exchange between these organisms could generate new virulence phenotypes with potentially severe consequences. The *Leishmania* parasites mate in the sand fly vector and also in vitro in culture, facilitating experimental analysis. The Sacks lab is at the forefront of research into genetic exchange in *Leishmania* and leads the field. In this paper they attempt to provide evidence of the mechanism of mating by deleting 7 genes, 5 involved in meiosis and 2 in cell fusion. In the first set of experiments using in vitro crosses, the readout was numbers of hybrids produced. Deletion of HOP1 or HAP2-2 severely disrupted production of hybrids, whereas deletion of other genes produced only slight or moderate effect. Production of hybrids could be restored by add-back of the deleted gene. The negative effects of deletion of HOP1 or HAP2-2 on mating were subsequently confirmed in the in vivo sand fly system. The two parental *Leishmania* strains showed a difference in dependence on HOP1 and HAP2-2 for successful mating, with a few hybrids being produced in certain crosses. Examination of these hybrids revealed that they were mostly triploid or tetraploid rather than diploid, indicating that meiosis and fusion had been disrupted though not completely halted. Finally expression of HOP1 and HAP2-2 was visualized inside live cells using fusion proteins linked to fluorescent reporters. Expression of both genes was restricted to a subpopulation of cells, with HOP1 expressed in the cell nucleus whereas HAP2-2 was expressed throughout the cell body.

The results are interesting and add to understanding of genetic exchange in *Leishmania*. However, I have some reservations about interpretation. The authors conclude that HOP1 and HAP2-2 are essential for genetic exchange in *Leishmania*, and by implication that genetic exchange involves a meiotic division and fusion of gametes. In the case of the MA37 parent, however, hybrids were produced despite deletion of HOP1 and HAP2-2, so these genes cannot be said to be essential for genetic exchange in *Leishmania*. The lack of significant depletion of hybrids with the 4 other meiosis genes tested is at odds with the clear disruption of hybrid production caused by deletion of HOP1. This weakens the case for meiosis, though the outcome of deletion of meiosis-specific genes is inevitably hard to predict. In other systems, e.g. plants, deletion of meiosis-specific genes causes chromosomal abnormalities in the hybrid progeny because the meiotic machinery fails to correctly segregate the chromosomes, so this approach is doubly difficult in *Leishmania*, where polyploid and aneuploid progeny are the norm. As the authors explain in the introduction, it is not known for certain that meiosis is involved in hybrid production in *Leishmania*, so it is difficult to understand why deletion of one meiosis-related gene but not others perturbs hybrid production. Lastly, the expression studies of fluorescent fusion proteins do not describe any particular phenotype for cells expressing HOP1 and HAP2-2. African trypanosomes expressing HOP1 in the nucleus were replicating cells with 2 kinetoplasts and flagella, consistent with these cells undergoing (meiotic) division, but the *Leishmania* equivalent had a single nucleus and kinetoplast – worth a comment? Similarly trypanosomes expressing HAP2 were predominantly gametes and meiotic intermediates – could this apply here? In summary, these are intriguing results but clearly not the whole story yet.

Reviewer #2:

Remarks to the Author:

1) What are the noteworthy results?

The authors demonstrated that two genes HOP1 and HAP2 are essential parts in Leishmania mating process using strains without these genes via a CRISPR/Cas9 gene editing method, based on their previous experiments. They also performed various experiments to support and provide insights to their results.

2a) Will the work be of significance to the field and related fields?

The analysis can be applied to other Leishmania species and the findings are valuable to understand the mating process of Leishmania and Trypanosomatida in general.

2b) How does it compare to the established literature? If the work is not original, please provide relevant references.

The group has published some related papers on experimental mating process in vitro, and these results led to the current manuscript. The findings are original and detail results are provided, even though the current manuscript may be too short to discuss their results in detail.

•Does the work support the conclusions and claims, or is additional evidence needed?

Their results clearly support the main claims, and alternative interpretations might be possible but are highly unlikely. Additional evidence is not needed for this paper, though additional descriptions of the methods may be valuable to readers if the length limit permits.

•Are there any flaws in the data analysis, interpretation and conclusions? Do these prohibit publication or require revision?

I cannot come up with any major/minor flaws in their original analysis, interpretation and conclusions, which require major revisions. I am familiar with the topics but not an expert in the experimental details.

•Is the methodology sound? Does the work meet the expected standards in your field?
The methodology sound for the study.

•Is there enough detail provided in the methods for the work to be reproduced?

The methods were probably enough for the manuscript. I think the paper is short probably because of the length limitation of the publication. The details were crammed into short sentences, so it was difficult to read in some experimental sections. If it is possible to add more descriptions within the given page limit, it will be easier to understand the paper.

General comments on the manuscripts:

My questions are underlined and then the relevant lines are show below with their line numbers.

Add “respectively” in this line? It is difficult to read through for those who are not familiar with these genes.

47 This conclusion is reinforced by
48 the identification of Leishmania homologs for meiotic genes 19, and their expression by
49 promastigote stages recovered from sand flies 20, including the core meiotic genes SPO11, HOP1
50 and DMC1, involved in creating DNA double-strand breaks, homologous chromosome alignment
51 and recombination.

GEX1 has only be mentioned without detail and GEX1 was not further discussed in the text. So GEX1 here can be removed, without adding an extra abbreviation.

51 and recombination. Upregulated expression of the Leishmania homolog genes encoding the cell
52 and nuclear fusion proteins HAP2/GCS1 (HAPLESS 2/Generative Cell-Specific 1) and GEX1 has

Are there some reasons using lower letter for hop1, hop2, mnd1, dmc1 here?

58 in Arabidopsis 22. Furthermore, the fact that some sexually reproducing organisms, e.g. Drosophila
59 melanogaster, lack many meiotic homologs, including hop1, hop2, mnd1, dmc1, means that many

Are there any strong evidences that support this “parasexual process” proposed by the Sterkers, Y. 2014? The FISH method over estimate chromosomal copy number variability. The single cell genome sequence (SCGS) data do not corroborate the previous assumptions that all chromosomes are found with at least two somy states. (Negreira 2022). Are there any real experimental evidence support this hypothesis, other than their group? And could the authors clarify the further insight that the current manuscript provide on Leishmania parasexuality and the potential roles of HOP1 and a HAP2 in its parasexuality.

Gabriel H Negreira et al.

Nucleic Acids Research, Volume 50, Issue 1, 11 January 2022, Pages 293-305, <https://doi.org/10.1093/nar/gkab1203>

High throughput single-cell genome sequencing gives insights into the generation and evolution of mosaic aneuploidy in Leishmania donovani

66 The aneuploid organization of the Leishmania genome has been argued to
67 favor a parasexual process to explain their mode of genetic exchange, in which fusion of diploid
68 cells is followed by random chromosome loss during subsequent mitotic divisions 27.

Would it possible to define what “:” mean in the manuscript?

110 For gene targeting, we used as background MA37 and L747
111 T7RNAPol::SpCas9 (T7Cas9) cell lines 16, which stably express Cas9 and T7RNA polymerase to
112 drive sgRNA transcription 34.

Was sgRNA defined at its first appearance in the paper, though it was defined shortly after?

112 drive sgRNA transcription

The authors performed a statistical test, so it might be better to explicitly say “statistically significant” here. The changes may appear non-random at first glance.

any significant changes => any statistically significant changes

125 We did not observe any significant changes in promastigote growth when comparing null
126 mutants to their parental cell lines, nor in untreated and irradiated mutant lines (Supplementary
127 Fig. 2).

Could ‘x’ defined? There are many abbreviations in the manuscript and it is hard to keep track. MA37 eGFP-Neo x L747 mCherry-Sat, I think “+” is used in the sup tables.

136 was compared to the rates obtained in MA37 eGFP-Neo x L747 mCherry-Sat

This is very important, clear result. But is it possible to describe this positive result to those who are not familiar with flow cytometry with a few more lines? Is it correct to say that the high concentration of “a quadrant of higher signal concentration” match with a double mutant hybrid?

138 proteins, respectively, all positive wells showing growth in the double drug selection medium were
139 confirmed using flow cytometry (Fig. 2A).

Could this result specific to these strains?

166 The partial recovery of

167 hybridizing compatibility with the MA37 HAP2-2 null mutant line was surprising since the crosses
168 undertaken using the HAP2-2 null mutant and the L747 control line (Fig. 2B-C) indicated that
169 HAP2-2 expression in both parents was required for hybridization in vitro.

Overall, I found this section difficult to read through. Is it possible to make this section simpler for those who are not familiar with the topic?

171 **Genetic exchange in the sand fly requires HOP1 and HAP2-2.**

Could this line be rephrased more clearly?

180 Homogenized midguts from

181 individual flies were placed in a double drug selection medium, and the frequency of dissected
182 flies yielding a double drug resistant hybrid and confirmed by flow cytometry analysis was
183 determined.

Flow cytometry and DNA content:

The flow cytometry results provided are clean but are these results of flow cytometry and DNA content always clearly classified as 2, 3, 4, 5, 6 ploidy as the figures suggested? How many total samples were tested flow cytometry and how many were shown? Or just some count “n”s here are just rounded up number to make the analysis clean and readable? As far as I know, it is difficult to classify these ploidy values for some samples. What kind of criteria is used to determine ploidy? How ploidy estimation was calibrated?

213 to control crosses (MA37 eGFP-Neo x L747 mCherry-Sat, $p = 0.041$; Fig. 4D). The control hybrids
214 were mainly diploid ($n=30$), although triploid ($n=16$) and tetraploid ($n=1$) hybrids were also
215 observed (Supplementary Fig. 6), similar to prior observations regarding the frequency of
216 polyploid hybrids in vivo 9,11,18. For the MA37 $\Delta hop1$ eGFP-Neo crosses, the hybrids were either
217 triploids ($n=2$) or tetraploids (Fig. 4A and D).

Discussion section

Overall discussion start with general broad discussion of sexual reproduction, some of which may not directly related to the experiments performed in the manuscript. Is it possible to reconstruct the discussion, so that the discussion will proportionally reflect the experiments performed for the manuscript? I would prefer that the authors provide more detail discussion on their experiments.

Could some references cited here again to clarify what “visually well documented” really mean here?

281 For many microbial eukaryotes (protists), sexual reproduction, broadly defined by the admixture
282 of parental genomes and the generation of recombinant progeny, can be visually well
documented.

Are “the HOP1 null mutants” and “ the HOP1 deleted lines” the same or something different in these lines? It would be easier for the readers to use single expression if possible when there are so many mutants are discussed in the paper.

320 each parent 18. The severe mating defect displayed by the HOP1 null mutants i
326 to occur. Still, since the hybridizing potential of the HOP1 deleted lines

LtMA37/NEO originating from Jordan, and LtL747/HYG from Israel. What is the genetic diversity in these regions? Would it be possible to provide the strain origins in the text?

409 For this study we used two background cell lines previously reported: L. tropica MA37 T7Cas9-
410 Hyg and L. tropica L747 T7Cas9-Hyg

Are there any notable long, high copy number CNVs observed in any hybrids?

Are the definition of heterozygous SNPs (the allele frequencies were between 0.15-0.85) remain the same for any chromosome some status? I assume the detail definition is not critical for the manuscript but for lines with extreme high ploidy, the definition may need to be adjusted.

575 SNPs were considered heterozygous if the allele frequencies were between 0.15-0.85 and
576 homozygous if >0.85 for each genomic position

What is an approximate number of heterozygous SNPs? 373K homozygous SNPs are rather large number but is this consider to be typical genetic variations in one region? 396K SNP sites were identified in L don complex for example. I am asking this because the large genetic variation of two strains might be a key to genetic hybrid. Or simply, a hybrid could not be readily detected when the genetic variation among strains were two small...

581 inheritance circos plots were generated with 372,866 homozygous marker differences between L.
582 tropica MA37-GFP and L747-mCherry cell lines labeled in green and red, respectively

What is homology, % identify or similarity score of the genes HAP2-1 HAP2-2?

Fig. 1.
Protein domains. The letters on the domains are important in the manuscript but they are too small.

Fig. 2.

Fig. 2B and 2E. % Positive well of HAAP2-2 MA37 Null mutant and L747 Re-expressor is 0%. The value 0% is very important here. So is it possible to add some marker “x” representing 0% manually? This will make the already good figures more readable for the readers.

Fig. 6 and 7.

Nice pictures. Is It possible to show larger readable scale bars and letters so that they are more informative since the scale bars have different numbers.

Sup Fig. 1. What are A and B shown next to WT? I cannot see any description.

Sup Tab 2 (?)

HOP1 mutants: the values of the row in yellow seem wrong.

Crossings (Replicate 2)	Sand fly midguts			Ploidy					
	Total	Positive midguts	% positive midguts	diploid		triploid		tetraploid	
				Total	%	Total	%	Total	%
L tropica MA37 eGFP-Neo + L tropica L747 mCH-Sat	48	31	65.0	17.00	54.84	13.00	41.94	1.00	3.23
L tropica MA37 Δ hop1 eGFP-Neo + L tropica L747 mCH-Sat	37	4	10.8	0.00	0.00	0.00	0.00	4.00	100.00
L tropica MA37 eGFP-Neo + L tropica L747 Δ hop1 mCH-Sat	0	0	0.0	-	-	-	-	-	-
L tropica MA37 Δ hop1 eGFP-Neo + L tropica L747 Δ hop1 mCH-Sat	0	0	0.0	-	-	-	-	-	-
L tropica MA37 eGFP-Neo + L tropica L747 Δ hop1::HOP1-Sat	33	10	30.3	5.00	50.00	0.00	40.00	5.00	10.00
L tropica MA37 Δ hop1 eGFP-Neo + L tropica L747 Δ hop1::HOP1 (Sat)	31	2	6.5	0.00	0.00	0.00	0.00	2.00	100.00

Sup Tab 3: mean parental chromosomal contribution in parental lines and hybrids.

WGS sample	20_L747	20_MA37
MA37GFPWT	0	2.02
L747mChWT	3.02	0
LtHyb1	1.34	0.3

I noticed that LtHyb1 20_MA37 has 0.3 contribution. Does this suggest heterogeneous chromosome copy number contribution or some computational error? Or is this illustration of very even parental ploidy contribution to hybrid or a lack of mosaic aneuploidy in the hybrids.

Table 1.

This is a critical table but it was hard to see and understand the combination of lines used for hybrid experiments. Is it possible to reform the table 1 like shown below to visualize the experimental settings and results? I think table 1 can be separated into 3 tables.

Parental strains														freq	P-va
L. tropica MA37 eGFP-Neo							L. tropica L747 mCherry-Sat								
hop1	spo11	mnd1	hop2	dmc1	hap2-1	hap2-2	hop1	spo11	mnd1	hop2	dmc1	hap2-1	hap2-2		
														40/72 (55.6%)	-
							x								
x															
x							x								
								x							
	x														
	x							x							
		x							x						
		x							x						
										x					
				x											
				x							x				
					x							x			
					x							x			
						x							x		
						x								x	

END of my review.

Reviewer #3:

Remarks to the Author:

The manuscript presents strong evidence that two conserved genes involved in sexual reproduction - one involved in chromosome pairing in the synaptonemal complex and a second involved in gamete fusion - play roles in mating in *Leishmania*, a parasite system from a group in which sexual reproduction has been at least somewhat mysterious. The work builds on previous work by the Sacks group and others showing that apparently meiotic offspring can be generated in vivo (in the sandfly vector) and in vitro. Taken as a whole, this body of work has revolutionised our understanding of transmission genetics in this group, and this manuscript is another important step on that journey.

The most important issue is that I think there may be some problems with table 1. As far as I can see, the results here don't match the figures - for example, the MA37 fluorescent x L747 Hop1-KO line has 29% recovery in the table, while the 'reciprocal cross' of MA37 hop1-KO x L747 fluorescent has 5.6% recovery, while figure 1C shows the MA37 KO line with higher recovery than L747 KO line. I think there are similar problems elsewhere (e.g with hap-2-2) across the whole table. Either I don't understand something or something has gone wrong with completing the table. The p-values also seem surprising (for hop-1 the p-value for 4/72 recovery is much higher than for 21/72 compared to the control recovery of 40/72.. not impossible but unlikely and there doesn't seem to be a massive difference in variance visible on the figure that would explain it.

Suggestions for improvements:

line 139-140. How surprising is it that the other KO crosses do not show a phenotype? I would like to have seen something in the discussion about how widely conserved the function of these other genes are in other eukaryotes, and thus whether the absence of KO phenotypes in *Leishmania* tells us anything about how different crossing-over and gamete fusion might be in this system.

line 149: I missed much explanation for the different phenotypes of HOP-1 deletions in the two strains. This seems surprising if HOP-1 is a necessary and conserved part of the synaptonemal complex.. for example, is there any difference in expression of HOP-1 gene in these strains (RNA-seq data from these was published in Iantorno et al. mBio 8:e01393-17.)

Does chromosome 31 share the same pattern of inheritance 'bias' as the other chromosomes? As far as I can see nothing is said about this chromosome in the manuscript, and on figure 4E it looks triploid, which would be _ i think _ quite surprising in *Leishmania*. Does the pipeline used here somehow 'normalise away' the unusual ploidy of chr 31? IF so, does it also do this for other chromosomes?

Some minor text things:

lines 16-17. Is 'implicated' strong enough to describe the experimental hybrid work?

line 18. 'using in vitro and in vivo hybridization protocols ' might be a bit unclear to someone not familiar with the previous literature. how about "hybrids generated from experimental selection of drug resistance markers both in vitro and in vivo (during sandfly infections)" or something like that.

line 22-23, lines 74 - I think a bit more could be said about what is known in *T. brucei* - I'm thinking of the work from Wendy Gibson and colleagues for example, I think HAP2 expression in gametes has been shown in that system, and that HOP1 is expressed early in gametogenesis.

line 43: 'naturally confined' - do we really know that? It might be appropriate to say something like "assumed to be".

figure 1- would it be useful for the reader to also show some cartoon of the canonical domain

organisation of the homologs of these proteins in model organisms, as well as the Leishmania proteins?

line 127 - was there any difference in morphology of cells during proliferation?

line 144: I struggled a tiny bit with the sentence: "For crosses involving the HOP1 null mutants, deletion of the gene only in the L747 parent was sufficient to profoundly impair hybridization, while deletion in only the MA37 parent did have a significant effect." Something is up with the 'while' here.. what is the contrast you are making? Would it be better written as 'while deletion in only the MA37 parent had a smaller but significant effect'.

line 167: Do you have any explanation for the only partial recovery of the phenotype here? Was there robust expression of HAP2-2 in the re-expression line?

line 207: This clause could be much more nicely written: "MA37 parent was deleted or not of HOP1 or HAP2-2,"

line 281-286: the first 6 or 7 lines of the discussion seems unnecessarily repetitive of introduction, and could be cut to a single sentence or two.

line 332 - I wasn't completely clear what 'their' was referring to here.

line 473: Cell lines generation - would be more grammatical as 'generation of cell lines'.

Figure 2 legend: meiotic-related genes -> meiosis-related genes

Reviewer #1

The topic of sexual reproduction in *Leishmania* is an important one because this genus includes several human pathogens with devastating global impacts. Genetic exchange between these organisms could generate new virulence phenotypes with potentially severe consequences. The *Leishmania* parasites mate in the sand fly vector and also in vitro in culture, facilitating experimental analysis. The Sacks lab is at the forefront of research into genetic exchange in *Leishmania* and leads the field. In this paper they attempt to provide evidence of the mechanism of mating by deleting 7 genes, 5 involved in meiosis and 2 in cell fusion. In the first set of experiments using in vitro crosses, the readout was numbers of hybrids produced. Deletion of HOP1 or HAP2-2 severely disrupted production of hybrids, whereas deletion of other genes produced only slight or moderate effect. Production of hybrids could be restored by add-back of the deleted gene. The negative effects of deletion of HOP1 or HAP2-2 on mating were subsequently confirmed in the in vivo sand fly system. The two parental *Leishmania* strains showed a difference in dependence on HOP1 and HAP2-2 for successful mating, with a few hybrids being produced in certain crosses. Examination of these hybrids revealed that they were mostly triploid or tetraploid rather than diploid, indicating that meiosis and fusion had been disrupted though not completely halted. Finally expression of HOP1 and HAP2-2 was visualized inside live cells using fusion proteins linked to fluorescent reporters. Expression of both genes was restricted to a subpopulation of cells, with HOP1 expressed in the cell nucleus whereas HAP2-2 was expressed throughout the cell body.

The results are interesting and add to understanding of genetic exchange in *Leishmania*. However, I have some reservations about interpretation. The authors conclude that HOP1 and HAP2-2 are essential for genetic exchange in *Leishmania*, and by implication that genetic exchange involves a meiotic division and fusion of gametes. In the case of the MA37 parent, however, hybrids were produced despite deletion of HOP1 and HAP2-2, so these genes cannot be said to be essential for genetic exchange in *Leishmania*. The lack of significant depletion of hybrids with the 4 other meiosis genes tested is at odds with the clear disruption of hybrid production caused by deletion of HOP1. This weakens the case for meiosis, though the outcome of deletion of meiosis-specific genes is inevitably hard to predict. In other systems, e.g. plants, deletion of meiosis-specific genes causes chromosomal abnormalities in the hybrid progeny because the meiotic machinery fails to correctly segregate the chromosomes, so this approach is doubly difficult in *Leishmania*, where polyploid and aneuploid progeny are the norm. As the authors explain in the introduction, it is not known for certain that meiosis is involved in hybrid production in *Leishmania*, so it is difficult to understand why deletion of one meiosis-related gene but not others perturbs hybrid production. Lastly, the expression studies of fluorescent fusion proteins do not describe any particular phenotype for cells expressing HOP1 and HAP2-2. African trypanosomes expressing HOP1 in the nucleus were replicating cells with 2 kinetoplasts and flagella, consistent with these cells undergoing (meiotic) division, but the *Leishmania* equivalent had a single nucleus and kinetoplast – worth a comment? Similarly trypanosomes expressing HAP2 were predominantly gametes and meiotic intermediates – could this apply here? In summary, these are intriguing results but clearly not the whole story yet.

Response to Reviewer #1:

We thank the reviewer for the thoughtful comments. We are in full agreement that these results are not the whole story yet. We do, however, believe that the HOP1 data strongly advances the case for meiosis. The few hybrids generated when the MA37 parent was deleted of HOP1 were all either triploid or tetraploid, indicating that while the mutant cells can still fuse, they fail to undergo reductional division, leading to the sort of chromosome abnormalities in the hybrid progeny that the reviewer referred to. We have expanded on this point in the discussion (lines 326-331):

“While a few hybrids could be recovered when the MA37 *HOP1* null mutant was crossed with L747 wild type, the products were all polyploid, with the deleted parent shown to be the source of the extra genome in the triploid hybrids. Thus, HOP1 in MA37 is required for reductional division even if it is not absolutely essential for hybridization to occur. These results are similar to the outcome of deletion of meiosis-specific genes in plants, for which defects in chromosome segregation can produce hybrid progeny with aneuploid or polyploid genomes (Hyde et al., Plant Biotechnol J, 2023).”

With regard to the lack of dependency on the other core meiotic genes tested for hybridization *in vitro*, we provide some discussion of this interesting point (lines 347-356):

“We also investigated the importance of other meiotic gene homologs involved in early events of meiosis I, namely SPO11, involved in DSB formation, and DMC1 and its accessory proteins MND1 and HOP2, involved in DNA strand invasion and recombination. We could not associate any of these genes to hybridization *in vitro*. In contrast to the effects of HOP1 deletion, the early stages of gametogenesis may be able to proceed in the absence of these axial-elements, or else there are proteins/pathways that provide redundant functions. In *C. elegans*, for example, radiation induced DSBs were shown to alleviate the requirement for SPO11⁴⁶. The RecA homologs DMC1 and RAD51 serve complementary functions for strand exchange between homologous chromosomes in *S. cerevisiae*, although homology search and strand invasion can still occur with low efficiency in the absence of DMC1⁴⁷.”

Finally, we have added some discussion of the published *T. brucei* data, drawing comparisons with *L. tropica* regarding the nature of HAP2 and HOP1 expression observed in the respective insect forms of the parasites (lines 404-414):

“In *T. brucei*, HAP2 expression is concentrated in some gametes at the cell posterior, while in other gametes and meiotic intermediates, it is expressed throughout the cytoplasm (Peacock, Kay et al., *Comm Biol.*, 2021). Whether it needs to be expressed on both gametes is not known. HAP2 expression in *T. brucei* is confined to life cycle stages in the tsetse salivary glands where gametes and meiotic intermediates are found. Similarly, HOP1 expression in *T. brucei* is restricted to cells in the salivary glands undergoing meiotic division, marked by the presence of two kinetoplasts (Peacock et al., *PNAS*, 2011). By comparison, the HAP2-2 expressing cells in *L. tropica* were distributed throughout the posterior and anterior midgut, including the plug, while the HOP1 expressing cells were mainly in the posterior midgut and did not show signs of cell division. We so far lack evidence that HAP2-2 and/or HOP1 expression is restricted to gametes or cells committed to a meiotic program.”

Reviewer #2:

1) What are the noteworthy results?

The authors demonstrated that two genes HOP1 and HAP2 are essential parts in *Leishmania* mating process using strains without these genes via a CRISPR/Cas9 gene editing method, based on their previous experiments. They also performed various experiments to support and provide insights to their results.

2a) Will the work be of significance to the field and related fields?

The analysis can be applied to other *Leishmania* species and the findings are valuable to understand the mating process of *Leishmania* and *Trypanosomatida* in general.

2b) How does it compare to the established literature? If the work is not original, please provide relevant references.

The group has published some related papers on experimental mating process in vitro, and these results led to the current manuscript. The findings are original and detail results are provided, even though the current manuscript may be too short to discuss their results in detail.

Does the work support the conclusions and claims, or is additional evidence needed?

Their results clearly support the main claims, and alternative interpretations might be possible but are highly unlikely. Additional evidence is not needed for this paper, though additional descriptions of the methods may be valuable to readers if the length limit permits.

Are there any flaws in the data analysis, interpretation and conclusions? Do these prohibit publication or require revision?

I cannot come up with any major/minor flaws in their original analysis, interpretation and conclusions, which require major revisions. I am familiar with the topics but not an expert in the experimental details.

Is the methodology sound? Does the work meet the expected standards in your field? The methodology sound for the study.

Is there enough detail provided in the methods for the work to be reproduced?

The methods were probably enough for the manuscript. I think the paper is short probably because of the length limitation of the publication. The details were crammed into short sentences, so it was difficult to read in some experimental sections. If it is possible to add more descriptions within the given page limit, it will be easier to understand the paper.

Responses to Reviewer #2:

General comments on the manuscripts:

My questions are underlined and then the relevant lines are show below with their line numbers.

Add "respectively" in this line? It is difficult to read through for those who are not familiar with these genes.

47 This conclusion is reinforced by

48 the identification of *Leishmania* homologs for meiotic genes 19, and their expression by
49 promastigote stages recovered from sand flies 20, including the core meiotic genes SPO11,
HOP1 50 and DMC1, involved in creating DNA double-strand breaks, homologous chromosome
alignment 51 and recombination.

R: We added ‘respectively’ at the end of the sentence for clarity (line 51).

GEX1 has only be mentioned without detail and GEX1 was not further discussed in the text. So GEX1 here can be removed, without adding an extra abbreviation.

51 and recombination. Upregulated expression of the Leishmania homolog genes encoding the cell 52 and nuclear fusion proteins HAP2/GCS1 (HAPLESS 2/Generative Cell-Specific 1) and GEX1 has

R: We removed reference to GEX1 from the text and adjusted the text accordingly.

Are there some reasons using lower letter for hop1, hop2, mnd1, dmc1 here?

58 in Arabidopsis 22. Furthermore, the fact that some sexually reproducing organisms, e.g. Drosophila

59 melanogaster, lack many meiotic homologs, including hop1, hop2, mnd1, dmc1, means that many

R: The gene names are now consistently written in capital letters.

Are there any strong evidences that support this “parasexual process” proposed by the Sterkers, Y. 2014? The FISH method over estimate chromosomal copy number variability. The single cell genome sequence (SCGS) data do not corroborate the previous assumptions that all chromosomes are found with at least two somy states. (Negreira 2022). Are there any real experimental evidence support this hypothesis, other than their group? And could the authors clarify the further insight that the current manuscript provide on Leishmania parasexuality and the potential roles of HOP1 and a HAP2 in its parasexuality.

66 The aneuploid organization of the Leishmania genome has been argued to

67 favor a parasexual process to explain their mode of genetic exchange, in which fusion of diploid
68 cells is followed by random chromosome loss during subsequent mitotic divisions 27.

R: We agree with the reviewer that the possible role of parasexuality in *Leishmania* has been overstated, as there is no data to directly support such a mode of genetic exchange. The argument that the meiotic machinery cannot recombine and segregate aneuploid genomes is incorrect - this is well described in plants and fungi. To the reviewers point that the Negreira paper has shown that many chromosomes remain consistently disomic, then the extent of mosaic aneuploidy is itself overstated. We have decided to remove the reference to parasexuality from the text.

Would it possible to define what “:” mean in the manuscript?

110 For gene targeting, we used as background MA37 and L747

111 T7RNAPol::SpCas9 (T7Cas9) cell lines 16, which stably express Cas9 and T7RNA
polymerase to

112 drive sgRNA transcription 34.

R: The name of the cell line was used as first described by Beneke et al., 2017, for which the plasmid was generated for *L. major* and *L. mexicana*. In general the double colon (::) indicates a double strand break and reunion. Here, it means both genes are expressed from the same plasmid, although not fused, because SpCas9 was inserted in a plasmid already used for expression of T7 RNA pol. To avoid confusion, we will refer to the cell line only as T7Cas9 (line 118).

Was sgRNA defined at its first appearance in the paper, though it was defined shortly after?
112 drive sgRNA transcription

R: We now define sgRNA (single guide RNA) at its first appearance in lines 118-119.

The authors performed a statistical test, so it might be better to explicitly say “statistically significant” here. The changes may appear non-random at first glance.
any significant changes => any statistically significant changes

125 We did not observe any significant changes in promastigote growth when comparing null
126 mutants to their parental cell lines, nor in untreated and irradiated mutant lines
(Supplementary
127 Fig. 2).

R: The text was modified as requested.

Could ‘x’ defined? There are many abbreviations in the manuscript and it is hard to keep track.
MA37 eGFP-Neo x L747 mCherry-Sat, I think “+” is used in the sup tables.

136 was compared to the rates obtained in MA37 eGFP-Neo x L747 mCherry-Sat

R: The use ‘x’ or ‘+’ indicating the crosses between two strains has now been replaced by the word ‘and’ in the text and tables.

This is very important, clear result. But is it possible to describe this positive result to those who are not familiar with flow cytometry with a few more lines? Is it correct to say that the high concentration of “a quadrant of higher signal concentration” match with a double mutant hybrid?

138 proteins, respectively, all positive wells showing growth in the double drug selection medium
were
139 confirmed using flow cytometry (Fig. 2A).

R: We have rephrased the description of the flow cytometry data (lines 146).

Could this result specific to these strains?

166 The partial recovery of
167 hybridizing compatibility with the MA37 HAP2-2 null mutant line was surprising since the
crosses

168 undertaken using the HAP2-2 null mutant and the L747 control line (Fig. 2B-C) indicated that
169 HAP2-2 expression in both parents was required for hybridization in vitro.

R: This of course remains a possibility. As we have only generated the mutant lines in the two *L. tropica* strains described, we can only infer the applicability of the findings to other species and strains.

Overall, I found this section difficult to read through. Is it possible to make this section simpler for those who are not familiar with the topic?

171 **Genetic exchange in the sand fly requires HOP1 and HAP2-2.**

R: The paragraph has been rewritten for clarity (lines 185-194).

Could this line be rephrased more clearly?

180 Homogenized midguts from
181 individual flies were placed in a double drug selection medium, and the frequency of dissected
182 flies yielding a double drug resistant hybrid and confirmed by flow cytometry analysis was
183 determined.

R: We have rephrased the description of the hybrid selection (lines 188-192).

Flow cytometry and DNA content:

The flow cytometry results provided are clean but are these results of flow cytometry and DNA content always clearly classified as 2, 3, 4, 5, 6 ploidy as the figures suggested? How many total samples were tested flow cytometry and how many were shown? Or just some count “n”s here are just rounded up number to make the analysis clean and readable? As far as I know, it is difficult to classify these ploidy values for some samples. What kind of criteria is used to determine ploidy? How ploidy estimation was calibrated?

213 to control crosses (MA37 eGFP-Neo x L747 mCherry-Sat, $p = 0.041$; Fig. 4D). The control
hybrids

214 were mainly diploid ($n=30$), although triploid ($n=16$) and tetraploid ($n=1$) hybrids were also
215 observed (Supplementary Fig. 6), similar to prior observations regarding the frequency of
216 polyploid hybrids in vivo 9,11,18. For the MA37 $\Delta hop1$ eGFP-Neo crosses, the hybrids were
either

217 triploids ($n=2$) or tetraploids (Fig. 4A and D).

R: DNA content assessed by flow cytometry was done for all hybrids recovered from sand fly infections, and results for ploidy are presented in figures 4 and 5 (hybrids from mutants' crossings), supplemental figure 6 (control hybrids) and summarized in Supplementary table 2. We used the parental diploid cell lines and previously confirmed diploid and tetraploid hybrids as references. Diploid hybrids were identified when their first peak in DNA content was the same as the first peak of the diploid controls, corresponding to G1 cells; tetraploid hybrids were identified when their first peak in DNA content was close to the second peak of

the diploid controls, corresponding to cells in G2/M. Cells with a DNA content of $3n$ were identified when their first peak localized in the valley (S-phase) between peaks G1 and G2/M from diploid controls. Because of mosaic aneuploidy in the cultured cells, their ploidy will not be integral, and hybrids clones that are each close to $2n$, for example, can show small differences in DNA content. We understand the reviewer's comment regarding the difficulty to predict ploidy, but this is an established method previously used by us and others (Cruz et al., 1993; Inbar et al., 2013; Louradour et al., 2020; Louradour, Ferreira et al., 2022). A few of the hybrids also had their ploidy assessed by WGS (please see Supplementary table 3 for parental contribution calculations, where the sum should predict total ploidy). The text was modified to better explain the analysis (lines 218-221).

Discussion section

Overall discussion start with general broad discussion of sexual reproduction, some of which may not directly related to the experiments performed in the manuscript. Is it possible to reconstruct the discussion, so that the discussion will proportionally reflect the experiments performed for the manuscript? I would prefer that the authors provide more detail discussion on their experiments. Could some references cited here again to clarify what "visually well documented" really mean here?

281 For many microbial eukaryotes (protists), sexual reproduction, broadly defined by the admixture

282 of parental genomes and the generation of recombinant progeny, can be visually well documented.

R: We have removed the more general points from the first paragraph of the discussion, including reference to "visually well documented" sex.

Are "the HOP1 null mutants" and "the HOP1 deleted lines" the same or something different in these

lines? It would be easier for the readers to use single expression if possible when there are so many mutants are discussed in the paper.

320 each parent 18. The severe mating defect displayed by the HOP1 null mutants i

326 to occur. Still, since the hybridizing potential of the HOP1 deleted lines

R: We feel that both descriptions are clear and interchangeable.

LtMA37/NEO originating from Jordan, and LtL747/HYG from Israel. What is the genetic diversity in

these regions? Would it be possible to provide the strain origins in the text?

409 For this study we used two background cell lines previously reported: *L. tropica* MA37 T7Cas9-

410 Hyg and *L. tropica* L747 T7Cas9-Hyg

R: We have previously reported high intraspecific allelic diversity within the *L. tropica* strains in the region, with at least 4 distinct ancestries identified across the 5 isolates from Israel and Jordan that were analyzed (Iantorno et al., mBio, 2017). We have provided the strain origins and WHO reference numbers in the text (lines 427-428).

Are there any notable long, high copy number CNVs observed in any hybrids?

R: We have not run that particular analysis which we feel is beyond the scope of the paper. The purpose of the WGS analysis was to inform genome wide inheritance patterns in the hybrids based on the parental contribution of the SNPs that were homozygous different between the parents.

Are the definition of heterozygous SNPs (the allele frequencies were between 0.15-0.85) remain the same for any chromosome copy status? I assume the detail definition is not critical for the manuscript but for lines with extreme high ploidy, the definition may need to be adjusted.

575 SNPs were considered heterozygous if the allele frequencies were between 0.15-0.85 and 576 homozygous if >0.85 for each genomic position

R: The SNP calling parameters were the same for all chromosomes. Based on our experience and the high allele frequency variability in *Leishmania* due to mosaic aneuploidy, those filters have been applied in other studies (e.g., PMID: 31091230, PMID: 34994687) to increase the chances of finding the highest number of heterozygous SNPs possible in the hybrid progeny while considering the variations in copy across and within cell populations. Allele frequencies of chromosomes that are trisomic are in the order of 0.33:0.67 and tetrasomic can reach the order of 0.2:0.8. This approach has been successfully applied by us for hybrid genomes with a ploidy up to $4n$ as it is the highest DNA content we have described in a sequenced genome so far.

What is an approximate number of heterozygous SNPs? 373K homozygous SNPs are rather large number but is this considered to be typical genetic variations in one region? 396K SNP sites were identified in L don complex for example. I am asking this because the large genetic variation of two strains might be a key to genetic hybrid. Or simply, a hybrid could not be readily detected when the genetic variation among strains were too small...

581 inheritance circles plots were generated with 372,866 homozygous marker differences between L.

582 *tropica* MA37-GFP and L747-mCherry cell lines labeled in green and red, respectively

R: The number of SNPs for each parental line and hybrids compared to the reference L590 genome are presented in Supplemental Table 3 (Tab 1). Briefly, we detected around 320K and 170K homozygous SNPs and 23K and 18K heterozygous SNPs in L747 and MA37, respectively. The homozygous differences were selected from the sum of the homozygous SNPs found in either one.

Among other reasons (e.g., high mating capacity), MA37 and L747 were chosen for this work due to their highly divergent genomes. *L. tropica* strains and isolates described so far are

known for their unusually high genetic plasticity and number of SNPs, in the order of ~300K according to other studies (PMID: 34968388 and PMID: 28900023).

What is homology, % identify or similarity score of the genes HAP2-1 HAP2-2?

R: We have provided the identity within the GCS1 domain (26.36%; line 108). The rest of the protein is not well conserved and the values for the full protein are misleading.

Fig. 1.

Protein domains. The letters on the domains are important in the manuscript but they are too small.

R: The requested modifications have been made. Please see the new Fig.1.

Fig. 2.

Fig. 2B and 2E. % Positive well of HAAP2-2 MA37 Null mutant and L747 Re-expressor is 0%. The value 0% is very important here. So is it possible to add some marker “x” representing 0% manually? This will make the already good figures more readable for the readers.

R: The requested modifications have been made. Please see the new Fig.2.

Fig. 6 and 7.

Nice pictures. Is it possible to show larger readable scale bars and letters so that they are more informative since the scale bars have different numbers.

R: The requested modifications have been made. Please see the new Fig.6 and 7.

Sup Fig. 1. What are A and B shown next to WT? I cannot see any description.

R: We added the information to the figure legend (line 916). The letters (A,B,C) on top of each lane represent a clone tested by diagnostic PCR.

Sup Tab 2 (?)

HOP1 mutants: the values of the row in yellow seem wrong.

R: Thank you for spotting this error. We corrected the value to 0.

Sup Tab 3: mean parental chromosomal contribution in parental lines and hybrids.

WGS sample 20_L747 20_MA37
MA37GFPWT 0 2.02
L747mChWT 3.02 0
LtHyb1 1.34 0.3

I noticed that LtHyb1 20 MA37 has 0.3 contribution. Does this suggest heterogeneous chromosome copy number contribution or some computational error? Or is this illustration of very even parental ploidy contribution to hybrid or a lack of mosaic aneuploidy in the hybrids.

R: This suggests heterogeneous parental contributions for this particular chromosome, as shown in the circus plot for each of the markers on chromosome 20 in LtHyb1 (Fig. 5D). The contribution of the extra chromosome from the L747 parent is not unexpected since the parent is trisomic for this chromosome (see Sup Table 3, Tab 2), and in a meiotic process would have a 50% chance of generating gametes with 2 copies.

Table 1.

This is a critical table but it was hard to see and understand the combination of lines used for hybrid experiments. Is it possible to reform the table 1 like shown below to visualize the experimental settings and results? I think table 1 can be separated into 3 tables.

R: The number of different recombinant lines used is indeed extensive. The listing of the parental lines involved in each crossing combination seems clear; we do not know how else to show this information in a table. We prefer to list all of the crossing combinations in a single table so that comparisons between the different experiments can better be made.

Reviewer #3

The manuscript presents strong evidence that two conserved genes involved in sexual reproduction - one involved in chromosome pairing in the synaptonemal complex and a second involved in gamete fusion – play roles in mating in *Leishmania*, a parasite system from a group in which sexual reproduction has been at least somewhat mysterious. The work builds on previous work by the Sacks group and others showing that apparently meiotic offspring can be generated in vivo (in the sandfly vector) and in vitro. Taken as a whole, this body of work has revolutionised our understanding of transmission genetics in this group, and this manuscript is another important step on that journey.

The most important issue is that I think there may be some problems with table 1. As far as I can see, the results here don't match the figures - for example, the MA37 fluorescent x L747 Hop1-KO line has 29% recovery in the table, while the "reciprocal cross" of MA37 hop1-KO x L747 fluorescent has 5.6% recovery, while figure 1C shows the MA37 KO line with higher recovery than L747 KO line. I think there are similar problems elsewhere (e.g with hap-2-2) across the whole table. Either I don't understand something or something has gone wrong with completing the table. The p-values also seem surprising (for hop-1 the p-value for 4/72 recovery is much higher than for 21/72 compared to the control recovery of 40/72.. not impossible but unlikely and there doesn't seem to be a massive difference in variance visible on the figure that would explain it.

R: Thank you for spotting this mistake, the values for crossings in MA37 and L747 mutants were swapped in Table 1 but correct in Supp. Table 1 and Fig. 1. We have now modified Table 1 to have the correct number, percentage of hybrids and p-value for each crossing.

Suggestions for improvements:

line 139-140. How surprising is it that the other KO crosses do not show a phenotype? I would like to have seen something in the discussion about how widely conserved the function of these other genes are in other eukaryotes, and thus whether the absence of KO phenotypes in *Leishmania* tells us anything about how different crossing-over and gamete fusion might be in this system.

R: We have addressed this interesting point in our response to Rev. 1, and in the discussion (lines 404-414).

line 149: I missed much explanation for the different phenotypes of HOP-1 deletions in the two strains. This seems surprising if HOP-1 is a necessary and conserved part of the synaptonemal complex.. for example, is there any difference in expression of HOP-1 gene in these strains (RNA-seq data from these was published in Iantorno et al. mBio 8:e01393-17.)

R: We also addressed this important point in our response to Rev 1 and in the discussion (lines 347-356).

Does chromosome 31 share the same pattern of inheritance 'bias' as the other chromosomes? As far as I can see nothing is said about this chromosome in the manuscript, and on figure 4E it looks triploid, which would be _ i think _ quite surprising in *Leishmania*. Does the pipeline used here somehow 'normalise away' the unusual ploidy of chr 31? IF so, does it also do this for other chromosomes?

R: Chromosome 31 shares the same inheritance bias as the other chromosomes and it has ~6 copies in hybrids LtHyb3-5, which explains why it looks trisomic on the circos plots. The values presented on Figs 4E and 5D indicate the frequencies of each parental homozygous difference in the hybrid progeny, not their somies. Therefore, some multiples of three should look similar to trisomic. Chromosome some values are presented in Supplemental File 3 along with each parental contribution.

Some minor text things:

lines 16-17. Is 'implicated' strong enough to describe the experimental hybrid work?

R: We rephrased the text to “strongly supported”.

line 18. 'using in vitro and in vivo hybridization protocols ' might be a bit unclear to someone not familiar with the previous literature. how about "hybrids generated from experimental selection of drug resistance markers both in vitro and in vivo (during sandfly infections)" or something like that.

R: We modified the text for clarity (lines 18-20).

line 22-23, lines 74 - I think a bit more could be said about what is known in *T. brucei* - I'm thinking of the work from Wendy Gibson and colleagues for example, I think HAP2 expression in gametes has been shown in that system, and that HOP1 is expressed early in gametogenesis.

R: Because of the strict word limitation, we do not refer to the *T. brucei* data in the abstract, which in any case to do not include functional studies. We have, however, expanded our discussion of the similarities and differences between *T. brucei* and *L. tropica* with regard to HAP2 and HOP1 expression in the stages present within their respective insect vectors (lines 404-414).

line 43: 'naturally confined' - do we really know that? It might be appropriate to say something like "assumed to be".

R: The text was modified as requested (line 43).

figure 1- would it be useful for the reader to also show some cartoon of the canonical domain organisation of the homologs of these proteins in model organisms, as well as the *Leishmania* proteins?

R: We modified the figure as requested and included in the text descriptions of the similarities and differences among the model organisms (lines 92-110)

line 127 - was there any difference in morphology of cells during proliferation?

R: We did not see any differences in morphology or motility of the null mutants or irradiated cells.

line 144: I struggled a tiny bit with the sentence: "For crosses involving the HOP1 null mutants, deletion of the gene only in the L747 parent was sufficient to profoundly impair hybridization, while deletion in only the MA37 parent did have a significant effect." Something is up with the 'while' here.. what is the contrast you are making? Would it be better written as 'while deletion in only the MA37 parent had a smaller but significant effect'.

R: The text has been modified for clarity (line 153).

line 167: Do you have any explanation for the only partial recovery of the phenotype here? Was there robust expression of HAP2-2 in the re-expression line?

R: Partial recovery of phenotypes following gene re-expression is a frequently observed phenomenon, not only in *Leishmania* but also in other organisms. This partial recovery can be attributed to several factors, one of which is the alteration of optimal protein levels, either by overexpression or insufficient expression when compared to endogenous wild-type levels.

line 207: This clause could be much more nicely written: "MA37 parent was deleted or not of HOP1 or HAP2-2,"

R: We rephrased the sentence (215-216).

line 281-286: the first 6 or 7 lines of the discussion seems unnecessarily repetitive of introduction, and could be cut to a single sentence or two.

R: The repetitive sentences in the first paragraph of the discussion have been deleted.

line 332 - I wasn't completely clear what 'their' was referring to here.

R: The text has been modified for clarity (line 350).

line 473: Cell lines generation - would be more grammatical as 'generation of cell lines'.

R: We modified the text as suggested (line 491).

Figure 2 legend: meiotic-related genes -> meiosis-related genes

R: We modified the text as requested (line 634).

Reviewers' Comments:

Reviewer #1:

Remarks to the Author:

The authors have satisfactorily addressed the comments made.

Reviewer #2:

Remarks to the Author:

The authors have addressed the most of the main concerns pointed out by the reviewers, and I believe the clarity of the paper was improved and the message and scope of the paper became cleaner. So I do not have any more major comments on the manuscript.

The last minor point which I do not need to see reply is as follows.

In "HOP mutants" table in 1436102_1_supp_8080614_s16rc8.xlsx

Please check this line. I believe the last value should be 50 not 10, if I understand the table correctly. This was supposedly already corrected.

L tropica MA37 eGFP-Neo + L tropica L747 Δ hop1::HOP1-Sat 33 10 30.3 5.00 50.00 0.00 0.00 5.00
10=>50

Reviewer #3:

Remarks to the Author:

The authors have addressed all my queries about the previous version of the manuscript and I'm happy to recommend publication

Point by point response:

Reviewer #1 (Remarks to the Author):

The authors have satisfactorily addressed the comments made.

Reviewer #2 (Remarks to the Author):

The authors have addressed the most of the main concerns pointed out by the reviewers, and I believe the clarity of the paper was improved and the message and scope of the paper became cleaner. So I do not have any more major comments on the manuscript.

The last minor point which I do not need to see reply is as follows.

In "HOP mutants" table in 1436102_1_supp_8080614_s16rc8.xlsx

Please check this line. I believe the last value should be 50 not 10, if I understand the table correctly. This was supposedly already corrected.

L tropica MA37 eGFP-Neo + L tropica L747 Δ hop1::HOP1-Sat 33 10 30.3 5.00 50.00 0.00 0.00 5.00
10=>50

The value in supplementary table 2 has been corrected to read 50%.

Reviewer #3 (Remarks to the Author):

The authors have addressed all my queries about the previous version of the manuscript and I'm happy to recommend publication